# From Token to Token Pair: Efficient Prompt Compression for Large Language Models in Clinical Prediction

**Mingcheng Zhu** [1]  **Zhiyao Luo** [1]  **Yu Liu** [1]  **Tingting Zhu** [1]

## Abstract

By processing electronic health records (EHRs) as natural language sequences, large language models (LLMs) have shown potential in clinical prediction tasks such as mortality prediction and phenotyping. However, longitudinal or highly frequent EHRs often yield excessively long token sequences that result in high computational costs and even reduced performance. Existing solutions either add modules for compression or remove less important tokens, which introduce additional inference latency or risk losing clinical information. To achieve lossless compression of token sequences without additional cost or loss of performance, we propose **Med**ical **T**oken-**P**air **E**ncoding (MedTPE), a layered method that extends standard tokenisation for EHR sequences. MedTPE merges frequently co-occurring medical token pairs into composite tokens, providing lossless compression while preserving the computational complexity through a dependency-aware replacement strategy. Only the embeddings of the newly introduced tokens of merely 0.5-1.0% of the LLM's parameters are fine-tuned via self-supervised learning. Experiments on real-world datasets for two clinical scenarios demonstrate that MedTPE reduces input token length by up to 31% and inference latency by 34-63%, while maintaining or even improving both predictive performance and output format compliance across multiple LLMs and four clinical prediction tasks. Furthermore, MedTPE demonstrates robustness across different input context lengths and generalisability to scientific and financial domains and different languages. The code is available in the [GitHub repository](#).

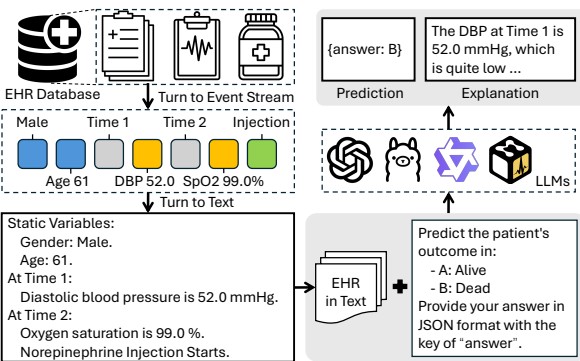

*Figure 1.* Illustration of the LLM-based clinical prediction.

[1]Department of Engineering Science, University of Oxford, Oxford, United Kingdom. Correspondence to: Mingcheng Zhu <mingcheng.zhu@eng.ox.ac.uk>.

*Proceedings of the 43$^{rd}$ International Conference on Machine Learning*, Seoul, South Korea. PMLR 306, 2026. Copyright 2026 by the author(s).

## 1. Introduction

Electronic health records (EHRs) document a longitudinal timeline of clinical events, including diagnosis, discharge notes, laboratory results, vital signs, medications, and procedures (Theodorou et al., 2023). By transforming these clinical events into natural language sequences, large language models (LLMs) can capture temporal and contextual patterns across the entire clinical trajectory, thereby supporting patient care and decision-making within healthcare systems (Lee et al., 2020; Zhu et al., 2026). Recent studies have reported that LLMs can perform a range of clinical prediction tasks in zero-shot settings and produce human-readable explanations, such as predictions of mortality and phenotyping (Renc et al., 2024; Cui et al., 2025; Williams et al., 2024). Specifically, this paradigm combines the EHR sequence in text with a task-specific prompt, allowing the LLM to generate both predictions and free-text explanations. This offers promising solutions for both predictive performance and interpretability, as illustrated in Figure 1.

While LLM-based clinical prediction seems promising, the transformation of longitudinal medical records often produces token sequences that exceed the context window limitations of most pre-trained LLMs (Wornow et al., 2025). For example, even a single stay in the intensive care unit (ICU) can result in a token sequence with length exceeding 64,000 due to the high frequency of clinical events (Fleming et al., 2024). Such lengthy sequences lead to increased computational requirements, slower inference, and limit

the feasibility of context-length and test-time scaling strategies that can improve the clinical prediction performance of LLMs. This inefficiency arises because widely used tokenisation algorithms, such as Byte-Pair Encoding (BPE), WordPiece, and SentencePiece (Sennrich et al., 2016; Song et al., 2021; Kudo & Richardson, 2018), were originally optimised for general language modelling and are not suitable for the complex and specialised vocabulary of clinical text. As a result, medical terms are fragmented into multiple subword tokens, unnecessarily extending the sequence length and computation (Yu, 2025). For example, a standard tokeniser divides the single clinical concept "Spirometry" into three separate tokens, [Spi', rom', 'etry'], rather than processing it as a unified term.

To address the challenge of long token sequences in EHRs, several approaches have been proposed, but each has notable limitations in the clinical context. One method is to develop a medical-specific vocabulary from medical corpora, which helps to avoid over-fragmented tokens (Bolton et al., 2024; Kim et al., 2024; Renc et al., 2024). However, this method requires the resource-intensive retraining of the entire LLM and may compromise the core capabilities of pre-trained LLMs. Removal-based compression methods, which discard less important tokens from the input, do not require model retraining, but risk omitting clinically significant information (Liskavets et al., 2025; Jiang et al., 2023; Pan et al., 2024). Merge-based compression methods, which dynamically merge medical tokens during inference, enable lossless prompt compression, but often introduce additional parameters or modules, thereby increasing inference latency and model complexity (Nakash et al., 2025; Han et al., 2025; Harvill et al., 2025). Consequently, there remains a need for a medical tokenisation approach that achieves lossless compression, maintains compatibility with pre-trained LLMs, and does not add further space or computational overhead.

To address the limitations of existing compression methods in the clinical context, we propose medical token-pair encoding (MedTPE), as illustrated in Figure 2. Specifically, MedTPE operates in three steps to achieve efficient lossless compression of token sequences in EHRs. First, it discovers and merges frequently co-occurring token pairs from EHR sequences, creating TPE tokens tailored to medical text. Next, MedTPE employs a dependency-aware replacement strategy, substituting approximately 3% of the least common original tokens in the pre-trained LLM's vocabulary with the most common TPE tokens. This strategy preserves the original vocabulary size and model parameters while ensuring the integrity of the original tokenisation process, retaining the same computational complexity as standard tokenisation methods. Finally, only the embeddings of the new TPE tokens are fine-tuned via self-supervised learning, while all other model parameters remain fixed. By design, MedTPE increases the information density of each token,

allowing for a more compact representation of the EHR sequence within the model's fixed context constraints. Overall, MedTPE delivers efficient and lossless compression that integrates smoothly with pre-trained LLMs, improving the inference efficiency of LLMs in clinical prediction tasks. Our main contributions are as follows:

- **Tokenisation-driven compression for clinical prediction.** We are the first to address the challenge of long token sequences of EHR in LLM-based clinical prediction by optimising the tokenisation process. The proposed method achieves lossless compression of EHR sequences, enhancing the efficiency of LLMs in clinical prediction across different tokenisers and LLM backbones.

- **Efficient and label-free tokenisation.** MedTPE preserves the computational complexity of standard tokenisation by maintaining the original tokenisation rules through the dependency-aware replacement. Furthermore, it fine-tunes the embeddings of new tokens with self-supervised learning, enabling embedding alignment without requiring any labelled data.

- **Compression without performance loss.** MedTPE achieves substantial compression while maintaining and even improving predictive performance on clinical tasks for the highly frequent and heterogeneous ICU scenario and the long and sparse longitudinal care. Beyond surpassing state-of-the-art compression strategies, MedTPE exhibits robustness across varying context lengths and demonstrates strong generalisability across clinical narratives, scientific reasoning, and financial summarisation.

## 2. Related Work

**LLM-based Prediction on EHRs**   Recent studies have leveraged LLMs for clinical prediction based on EHR (Chen et al., 2024a; Fleming et al., 2024; Niu et al., 2024; Wu et al., 2024). EHR-KnowGen (Niu et al., 2024) extracted targeted subsets of medical events and laboratory results from EHR sequences, presenting them as narrative summaries for input to the LLM. ClinicalBench (Chen et al., 2024a) converted diagnosis, procedure, and medication codes into descriptive sentences to enrich the clinical context available to the model. Similarly, Llemr (Wu et al., 2024) transformed the entire set of EHR events into descriptive sentences, embedding these events using ClinicalBERT (Alsentzer et al., 2019) before passing them onto the LLM. Moreover, MedAlign (Fleming et al., 2024) transformed complete patient event histories into XML-formatted text for LLM input. Despite their successes, these approaches encountered challenges with excessively long token sequences, which they addressed either by omitting portions of clinical events that risk losing important information or by employing hybrid encoding schemes that increase model complexity.

**Tokenisation and Compression Strategies** Modern LLMs commonly use subword tokenisation methods to represent rare or out-of-vocabulary words using a fixed-size vocabulary, such as BPE (Sennrich et al., 2016), WordPiece (Song et al., 2021), and SentencePiece (Kudo & Richardson, 2018). BPE iteratively merges frequent adjacent characters into subword tokens. WordPiece optimises merges based on language model predictability, while SentencePiece selects tokens through iterative probabilistic pruning. Although effective for general text, these tokenisers often over-segment specialised medical terms, leading to longer token sequences, higher computational costs, and reduced semantic cohesion (Hasan et al., 2024).

To address this, prompt compression methods have been proposed to reduce the input sequence length, generally falling into two categories: removal-based and merge-based. Removal-based methods evaluate the importance of individual tokens or sentences and selectively remove those deemed less relevant from the input sequence. Specifically, LLMLingua (Jiang et al., 2023) and LLMLingua2 (Pan et al., 2024) implement token-level compression by estimating token importance and discarding lower-ranked tokens. In contrast, CPC (Liskavets et al., 2025) operates at the sentence level, measuring the semantic relevance between each context sentence and the query, subsequently retaining only those most relevant to the given question. However, these methods risk discarding diagnostic nuances that are essential for clinical fidelity, potentially compromising the performance of clinical predictions.

Merge-based methods, conversely, create domain-specific tokens by aggregating frequent co-occurring units. For example, AdaptiVocab (Nakash et al., 2025) dynamically replaces less useful tokens with domain-specific ones during inference, thus increasing computational complexity. The meta-token method named LTSC (Harvill et al., 2025) replaces co-occurring tokens with a single meta-token, also requiring dynamic inference-time replacement. Both of these approaches require supervised alignment of the newly introduced embeddings with labelled data to ensure the model remains effective within the target domain. Similarly, ZeTT (Minixhofer et al., 2024) uses a hypernetwork to generate embeddings for new tokens by aggregating information from their original constituent sub-tokens, thereby extending both the original vocabulary and the embedding space. In contrast, our method maintains the original vocabulary size and model parameter count, preserves the computational complexity of the original tokenisation, and eliminates the need for labelled data using self-supervised learning. More information can be found in Table 1.

*Table 1.* Summary of prompt compression methods, describing methods that achieve lossless compression and label-free training, maintain their original vocabulary size and parameter count, and their tokenisation complexity (where $n$ is the sequence length).

| Method | Lossless | Label-free | Same Size | Complexity |
|---|---|---|---|---|
| LLMLingua | ✗ | ✓ | ✓ | $\mathcal{O}(n^2)$ |
| LLMLingua2 | ✗ | ✓ | ✓ | $\mathcal{O}(n^2)$ |
| CPC | ✗ | ✓ | ✓ | $\mathcal{O}(n^2)$ |
| AdaptiVocab | ✓ | ✗ | ✓ | $\mathcal{O}(n^2)$ |
| LTSC | ✓ | ✗ | ✗ | $\mathcal{O}(n \log n)$ |
| ZeTT | ✓ | ✓ | ✗ | $\mathcal{O}(n)$ |
| **MedTPE (Ours)** | ✓ | ✓ | ✓ | $\mathcal{O}(n)$ |

## 3. Preliminaries

Formally, we represent the longitudinal EHR for a patient $i \in \{1, \ldots, N_p\}$ as a sequence of timestamped events $\mathbf{E}^{(i)} = \{e_j^{(i)}\}_{j=1}^{T^{(i)}}$. Each event is defined as a tuple $e_j^{(i)} = (c_j^{(i)}, o_j^{(i)}, t_j^{(i)})$, comprising a clinical concept $c_j^{(i)} \in \mathcal{C}$ (e.g., diagnosis, medication code), an observation value $o_j^{(i)} \in \mathcal{O}$ (e.g., lab result), and a timestamp $t_j^{(i)} \in \mathbb{R}$. The sequence is ordered chronologically such that $t_j^{(i)} \leq t_{j+1}^{(i)}$, allowing multiple events to occur at the same timestamp. Subsequently, each structured event is transformed into a natural language sequence $s_j^{(i)} = \phi(e_j^{(i)})$ via a data-to-text function $\phi : \mathcal{C} \times \mathcal{O} \times \mathbb{R} \to \mathcal{S}$. These sequences are concatenated chronologically to form the full patient history:

$$S^{(i)} = s_1^{(i)} \oplus s_2^{(i)} \oplus \cdots \oplus s_{T^{(i)}}^{(i)}. \tag{1}$$

Finally, this aggregate text $S^{(i)}$ is processed by a tokeniser $\tau : \mathcal{S} \to \mathcal{V}^*$ to yield a discrete token sequence as

$$\mathcal{X}^{(i)} = \{x_n^{(i)}\}_{n=1}^{L^{(i)}}, \qquad x_n^{(i)} \in \mathcal{V}, \tag{2}$$

where $\mathcal{V}$ is the tokeniser vocabulary and $L^{(i)}$ is the sequence length. For clinical prediction, a natural language prompt $s_{\text{pmt}}$ (e.g., "What is the discharge diagnosis?") is tokenised into a sequence $\mathbf{P} = \{p_m\}_{m=1}^M$ and concatenated to form the final input:

$$\mathcal{X}'^{(i)} = \mathcal{X}^{(i)} \oplus \mathbf{P}. \tag{3}$$

A pre-trained autoregressive model $f_\theta$ generates the output sequence $\mathcal{G}^{(i)} = (g_1^{(i)}, \ldots, g_{K^{(i)}}^{(i)})$ by modelling the conditional probability:

$$p(\mathcal{G}^{(i)} \mid \mathcal{X}'^{(i)}; \theta) = \prod_{k=1}^{K^{(i)}} p(g_k^{(i)} \mid \mathcal{X}'^{(i)}, g_{<k}^{(i)}; \theta), \tag{4}$$

where $K^{(i)}$ is the output length. A task-specific extraction function $ext : \mathcal{V}^* \to \mathcal{Y}$ then maps the generated text to a clinical prediction:

$$\hat{y}^{(i)} = ext(\mathcal{G}^{(i)}). \tag{5}$$

In the following, we use $x$ to denote a generic token.

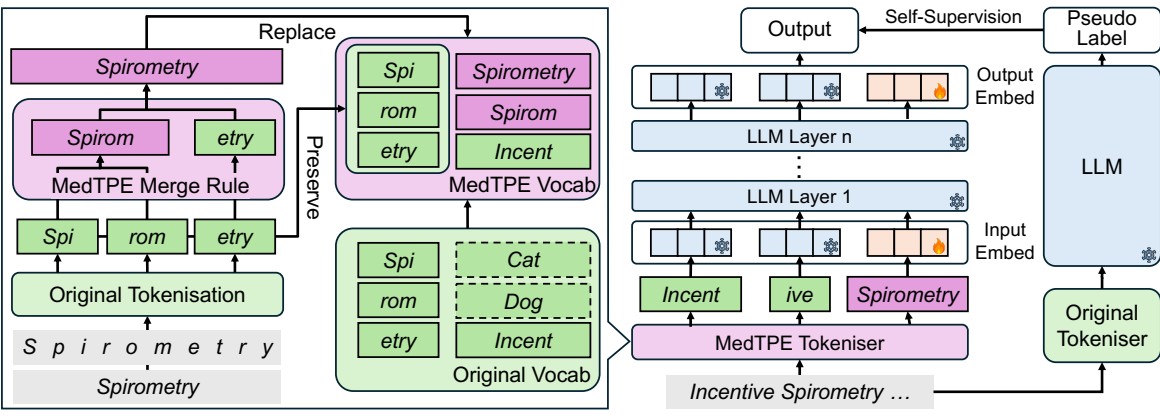

*Figure 2.* Overview of MedTPE tokenisation and its integration with LLMs. (a) **Token-pair encoding:** MedTPE identifies frequently co-occurring pairs in a medical corpus to form unified TPE tokens. (b) **Dependency-aware replacement:** The vocabulary is optimised by replacing low-utility general tokens (e.g., replacing "*Cat*" with "*Spirometry*") with high-value medical tokens, while strictly retaining all dependent sub-tokens to preserve the original tokenisation logic. (c) **Self-supervised fine-tuning (SSFT):** The original LLM processes the input ("*Incentive Spirometry*") to generate pseudo-labels. These labels supervise the fine-tuning of *only* the new TPE token embeddings, aligning them with the pre-trained latent space while the rest of the model remains frozen.

## 4. Methodology

The core design of MedTPE, as illustrated in Figure 2, consists of three modules that transform the original tokenisation into a medical-domain–optimised process. First, TPE mines a medical corpus to identify and merge frequently co-occurring original tokens into composite TPE tokens. Second, dependency-aware replacement integrates these TPE tokens into the original vocabulary while preserving all merge dependencies, thus keeping the vocabulary size and tokenisation complexity unchanged. Third, a light self-supervised fine-tuning (SSFT) step aligns the new TPE embeddings with the pre-trained embedding spaces so that the LLM can leverage the compressed, clinically relevant tokens for downstream prediction tasks.

### 4.1. Token-pair Encoding (TPE)

TPE is designed to encode subwords in medical terminology as single, semantically meaningful medical tokens, as illustrated in Figure 2(a). TPE functions as an extra layer built upon any base tokeniser, treating original subword units as the fundamental atoms for its merging operations.

**Vocabulary Discovery.** The construction of the TPE vocabulary, $\mathcal{V}_{\text{TPE}}$, begins with a data-driven discovery phase on a large-scale medical corpus. Initialising with the base tokeniser's vocabulary $\mathcal{V}$, we systematically identify contiguous sequences of $N$ tokens (where $2 \leq N \leq n_{\max}$) that represent cohesive clinical concepts. We calculate the global frequency of these $N$-gram candidates of TPE tokens in the medical corpus to identify a pool of clinical terms that are frequently fragmented by general-purpose tokenisers.

This candidate set serves as the basis for the subsequent dependency-aware replacement strategy.

**Encoding Process.** The TPE tokenisation process, $\tau^{\star}$, consists of two stages. First, the input text is processed by the base tokeniser to generate an intermediate sequence of original tokens. Subsequently, the TPE tokeniser applies a structured merge table, $\mathcal{M}_{\text{TPE}}$, to further consolidate these units. Specifically, for any contiguous span of length $N \in \{2, \ldots, n_{\max}\}$, a TPE token is constructed as:

$$d_j = x_1 \oplus x_2 \oplus \cdots \oplus x_N, \quad x_i \in \mathcal{V}, \quad d_j \in \mathcal{V}_{\text{TPE}}, \quad (6)$$

where $\oplus$ denotes string concatenation, and $x_i, x_{i+1}$ are adjacent sub-tokens.

To formalise the encoding of $d_j$ based on tokens $\{x_i\}_{i=1}^{N}$, we traverse its constituent sub-tokens from left to right to construct the merge path $MP(d_j)$:

$$MP(d_j) = \Big[ (x_1, x_2), \ (x_1 x_2, x_3), \ldots, (x_1 \cdots x_{N-1}, x_N) \Big]. \quad (7)$$

The TPE merging rule is defined by the merge table $\mathcal{M}_{\text{TPE}}$, which is generated by concatenating the unique merge paths of all discovered TPE tokens, ordered by their empirical frequency in the medical corpus:

$$\mathcal{M}_{\text{TPE}} = \big[ MP(d_1) \| MP(d_2) \| \ldots \| MP(d_J) \big], \quad (8)$$

where $\|$ denotes list concatenation with duplicate removal. This layered approach achieves substantial compression of EHR sequences while maintaining a computational complexity of $\mathcal{O}(n)$, where $n$ is the length of the input sequence. Crucially, this encoding process is information-lossless, ensuring that the original input text can be completely recon-

structed through decoding (detokenise($\tau^*(s)) = s$). Although the initial average embeddings are representationally lossy, the subsequent SSFT step recovers the representational fidelity required for high-precision clinical inference.

### 4.2. Dependency-aware Replacement

The dependency-aware replacement strategy is central to integrating the original and TPE vocabularies without increasing the overall vocabulary size while ensuring the layered TPE tokenisation. A straightforward solution is simply taking the union of the original and TPE vocabularies $\mathcal{V} \cup \mathcal{V}_{\text{TPE}}$, which would enlarge the embedding matrix and reduce computational efficiency. Instead, we maintain the original vocabulary size by replacing the least frequent original tokens with the most beneficial TPE tokens. To identify which TPE tokens to include, we use a length-aware frequency score that prioritises tokens that both occur frequently and contribute more to sequence compression:

$$\text{score}(d_j) = \text{freq}(d_j) \cdot |d_j|_{\text{orig}}, \tag{9}$$

where $\text{freq}(d_j)$ is the raw corpus frequency of token $d_j$, and $|d_j|_{\text{orig}}$ is the number of composited original tokens.

We select the top-$M$ TPE tokens ranked by this score to form the insertion set $\mathcal{I}$. Each TPE token $d_j \in \mathcal{I}$ is encoded with a specific merge path $MP(d_j)$ that starts from the bytes of input. To ensure correct layered tokenisation, we retain all original tokens involved in these merge paths, including any tokens that are required recursively along the merge path. This guarantees the presence of all necessary tokens for the original tokenisation. We refer to this preserved collection as the *dependent set* $\mathcal{D}$, defined as

$$\mathcal{D} = \left\{ x \; \middle| \; \begin{array}{l} \exists \, d_j \in \mathcal{I}, \exists \, u : (u, x) \in MP(d_j), \\ \text{or} \\ \exists \, (u, x') \in MP(d_j), \exists \, v : (v, x) \in MP(x') \end{array} \right\}. \tag{10}$$

Given the dependent set $\mathcal{D}$, we identify tokens eligible for replacement in the unprotected set $\mathcal{U} = \mathcal{V} \setminus \mathcal{D}$. We then select the $M$ least frequent tokens from $\mathcal{U}$ according to occurring frequency, forming the eviction set $\mathcal{E} \subseteq \mathcal{U}$. Combining removals and insertions gives the MedTPE vocabulary $\mathcal{V}^\star$:

$$\mathcal{V}^\star = (\mathcal{V} \setminus \mathcal{E}) \cup \mathcal{I}, \quad |\mathcal{I}| = |\mathcal{E}| = M. \tag{11}$$

For tokens in the eviction set $\mathcal{E}$ that are encountered during inference, the tokeniser returns to their underlying byte-level or sub-token representations present in the preserved vocabulary $\mathcal{V} \setminus \mathcal{E}$. This mechanism ensures that after removing rare tokens, the vocabulary remains computationally complete and capable of encoding any input string. This replacement strategy also prevents any increase in vocabulary size or embedding matrix, making MedTPE an efficient and compatible module for existing LLMs. The process is illustrated in Algorithm 1.

### 4.3. Self-supervised Fine-tuning (SSFT)

To align the embeddings of the newly introduced TPE tokens with the latent space of the pre-trained LLM, we employ SSFT. This process involves minimising the cross-entropy loss between the model's predictions and pseudo-labels generated by the original, uncompressed LLM. Specifically, we employ the original model to generate pseudo-labels for the training samples. The LLM, integrated with the MedTPE tokeniser, is then fine-tuned to predict these labels. This self-supervised alignment leverages the original model's generative behaviour as a supervisory signal, allowing the new token embeddings to seamlessly integrate into the pre-trained latent space without requiring manual annotations.

To facilitate training, we initialise the embedding of each new TPE token based on the embeddings of its constituent original tokens. For a TPE token $d_j$ composed of $N$ constituent original tokens $\{x_1, x_2, \cdots, x_N\}$, we initialise its embedding vector $\mathbf{e}_{d_j}$ by computing the arithmetic mean of the pre-trained embeddings of its constituent tokens:

$$\tilde{\mathbf{e}}_{d_j} = \frac{1}{N} \sum_{i=1}^{N} \mathbf{e}_{x_i}. \tag{12}$$

To maintain numerical stability, we normalise this vector to the weighted average norm of all original token embeddings. Specifically, the initial embedding is given by:

$$\mathbf{e}_{d_j} = \alpha \cdot \mu \cdot \frac{\tilde{\mathbf{e}}_{d_j}}{\|\tilde{\mathbf{e}}_{d_j}\|}, \quad \mu = \frac{1}{|\mathcal{V}|} \sum_{x \in \mathcal{V}} \|\mathbf{e}_x\|, \tag{13}$$

where $\mu$ denotes the average norm of embeddings across the BPE vocabulary $\mathcal{V}$. $\alpha$ is a scaling factor for normalisation, and we set $\alpha = 0.5$ in this study. Furthermore, to utilise the pre-trained LLM and improve the fine-tuning efficiency, we freeze all parameters of the LLM and train only the embeddings of the newly introduced TPE tokens. This self-supervised approach preserves the original capability of LLM while integrating the MedTPE tokeniser with LLM without labels. The process is illustrated in Algorithm 2.

## 5. Experiments

To evaluate MedTPE, we formulate the following key research questions (RQs): **RQ1 (Effectiveness)** assesses whether MedTPE outperforms existing prompt compression baselines; **RQ2 (Ablation)** isolates the contributions of token merging and embedding alignment; **RQ3 (Efficiency)** quantifies the setup costs; **RQ4 (Robustness)** examines performance stability across varying context lengths; and **RQ5 (Generalisability)** tests the method's transferability to non-clinical domains.

*Table 2.* Assessment of LLMs with MedTPE. Mean and standard deviation of F1 scores are reported, calculated by bootstrapping the test set 1,000 times. Inference time is reported in minutes, with the percentage change shown relative to the original LLM. **Bold** values indicate the best performance among the compression methods, and underlined values indicate the second-best performance.

*(a)* Comparison of prompt compression methods on MIMIC-IV

| Model | ICU Mortality | | | | Phenotyping | | | |
|---|---|---|---|---|---|---|---|---|
| | F1 ↑ | FCR ↑ | Time ↓ | CR↑ | F1 ↑ | FCR ↑ | Time ↓ | CR↑ |
| Llama3-1B | $0.033_{\pm0.007}$ | 0.330 | 66.0 | - | $0.050_{\pm0.002}$ | 0.492 | 50.2 | - |
| + T5Summary | $\underline{0.097}_{\pm0.005}$ | 0.872 | **18.3** | 99.1% | $0.078_{\pm0.003}$ | 0.838 | **13.6** | 98.9% |
| + LLMLingua2 | $0.035_{\pm0.003}$ | 0.328 | 82.5 | 32.1% | $0.058_{\pm0.003}$ | 0.500 | 38.7 | 32.4% |
| + ZeTT | $0.017_{\pm0.002}$ | **0.999** | 201.8 | 32.1% | $\underline{0.107}_{\pm0.005}$ | **0.997** | 113.2 | 32.4% |
| + MedTPE (Ours) | $\mathbf{0.109}_{\pm0.006}$ | 0.891 | 39.3 | 32.1% | $\mathbf{0.114}_{\pm0.004}$ | 0.870 | 22.3 | 32.4% |
| Qwen2.5-1.5B | $0.122_{\pm0.006}$ | 0.998 | 38.3 | - | $0.201_{\pm0.005}$ | 0.969 | 29.7 | - |
| + T5Summary | $0.100_{\pm0.006}$ | 0.886 | **6.1** | 99.1% | $0.073_{\pm0.004}$ | 0.865 | **7.7** | 98.8% |
| + LLMLingua2 | $\underline{0.103}_{\pm0.006}$ | 0.899 | 57.5 | 27.2% | $0.020_{\pm0.001}$ | 0.399 | 41.0 | 29.7% |
| + ZeTT | $0.001_{\pm0.001}$ | 0.751 | 194.2 | 27.2% | $\underline{0.112}_{\pm0.004}$ | 0.897 | 115.2 | 29.7% |
| + MedTPE(Ours) | $\mathbf{0.122}_{\pm0.006}$ | **1.000** | 23.5 | 27.2% | $\mathbf{0.218}_{\pm0.005}$ | **0.999** | 13.0 | 29.7% |

*(b)* Comparison of prompt compression methods on EHRSHOT

| Model | 30-day Readmission | | | | 1-year Pancreatic Cancer | | | |
|---|---|---|---|---|---|---|---|---|
| | F1 ↑ | FCR ↑ | Time ↓ | CR↑ | F1 ↑ | FCR ↑ | Time ↓ | CR↑ |
| Llama3-1B | $0.098_{\pm0.008}$ | 0.425 | 28.1 | – | $0.039_{\pm0.006}$ | 0.748 | 28.5 | – |
| + T5Summary | $\underline{0.190}_{\pm0.011}$ | 0.909 | **5.8** | 99.9% | $\underline{0.039}_{\pm0.007}$ | 0.940 | **2.5** | 99.9% |
| + LLMLingua2 | $0.071_{\pm0.008}$ | 0.326 | 57.8 | 27.5% | $0.024_{\pm0.005}$ | 0.503 | 39.3 | 28.3% |
| + ZeTT | $0.030_{\pm0.012}$ | **0.956** | 199.4 | 27.5% | $0.017_{\pm0.017}$ | **0.993** | 122.7 | 28.3% |
| + MedTPE(Ours) | $\mathbf{0.191}_{\pm0.011}$ | 0.877 | 16.6 | 27.5% | $\mathbf{0.047}_{\pm0.006}$ | 0.900 | 15.9 | 28.3% |
| Qwen2.5-1.5B | $0.205_{\pm0.011}$ | 0.989 | 15.2 | – | $0.047_{\pm0.010}$ | 0.877 | 30.8 | – |
| + T5Summary | $0.166_{\pm0.011}$ | 0.817 | **2.5** | 99.9% | $0.018_{\pm0.007}$ | 0.780 | **2.3** | 99.9% |
| + LLMLingua2 | $\underline{0.176}_{\pm0.012}$ | 0.860 | 25.3 | 22.8% | $\underline{0.044}_{\pm0.007}$ | 0.860 | 32.1 | 23.1% |
| + ZeTT | $0.002_{\pm0.002}$ | 0.614 | 117.9 | 22.8% | 0.000 | 0.884 | 120.6 | 23.1% |
| + MedTPE(Ours) | $\mathbf{0.209}_{\pm0.011}$ | **0.990** | 10.0 | 22.8% | $\mathbf{0.066}_{\pm0.017}$ | **0.947** | 11.5 | 23.1% |

## 5.1. Experiment Setup

**Datasets & Tasks** To evaluate the effectiveness of MedTPE, we conducted experiments on MIMIC-IV (Johnson et al., 2023) and EHRSHOT (Wornow et al., 2023). MIMIC-IV is an ICU dataset characterised by highly sampled data and a wide variety of clinical features of an ICU at the Beth Israel Deaconess Medical Centre in Boston. EHRSHOT includes patient sequences from 1990 to 2023 with irregular timestamps from Stanford Hospital, reflecting the longitudinal complexity of real-world scenarios. The statistics of the datasets are shown in Appendix Table 5. In MIMIC-IV, we have evaluated model performance on two clinical prediction tasks using the first 24 hours of data: (1) ICU Mortality, a binary classification task predicting whether a patient will die during their ICU stay, and (2) Phenotyping, a multi-label classification task predicting the presence of 25 clinical phenotypes. In EHRSHOT, we evaluate tasks (3) 30-day Hospital Readmission, predicting whether a patient will be readmitted within 30 days of discharge, and (4) 1-year Pancreatic Cancer Prediction, predicting pancreatic cancer diagnosis within one year.

**Metrics & Baselines.** We evaluate LLMs using four key metrics: the F1 score for predictive performance, the format compliance rate (FCR) for format adherence (Ni et al.,

2026), the inference time for inference efficiency, and the compression rate (CR) to quantify the reduction in the length of the token sequence. The LLMs evaluated include Qwen2.5 (Team, 2024), Llama3 (Grattafiori et al., 2024).

As primary baselines for prompt compression, we employ three distinct approaches: a specialised clinical text summarisation method T5Summary (Wilson et al., 2025), the removal-based compression method LLMLingua2 (Pan et al., 2024), and the merge-based compression method ZeTT (Minixhofer et al., 2024). To ensure a rigorous comparison, we configure ZeTT with the same vocabulary as MedTPE to eliminate the impact of different tokenisation granularities and align LLMLingua2's CR to match our method. More details can be found in Appendix C.

## 5.2. RQ1: Comparison of MedTPE with Other Methods

Table 2 presents the experimental results on MIMIC-IV and EHRSHOT. In both datasets, MedTPE reduces inference time by 33.9% to 62.5% and achieves compression rates between 22.8% and 32.4%. Compared to prompt compression baselines (LLMLingua2 and ZeTT), MedTPE consistently improves inference efficiency while preserving, and often enhancing, predictive performance across diverse LLMs and tasks. In particular, baseline methods often increase total in-

ference latency despite compressing the input. We attribute the specific underperformance of ZeTT to its reliance on a hypernetwork to generate embeddings. While effective for general vocabulary adaptation, hypernetworks struggle to capture the semantic meaning from the combination of subtokens (e.g., "hypotension" in the clinical context is quite different from "hypo" and "tension" in the general context). This semantic ambiguity forces the LLM to generate longer and more redundant output sequences to resolve the confusion, thereby extending generation time (Li et al., 2023). Compared to the summarisation baseline (T5Summary), MedTPE produces higher F1 scores. Although T5Summary achieves lower inference latency through aggressive compression, it compromises predictive performance, especially on challenging tasks like phenotyping prediction. MedTPE, in contrast, achieves a substantial gain in inference efficiency without sacrificing the reliability of clinical predictions. MedTPE consistently outperforms the baseline methods, even when they are augmented with an embedding fine-tuning step (Appendix H). Furthermore, MedTPE demonstrates robust effectiveness across a wide range of model families and parameter scales (1B to 32B), different training domains (Appendix E), and Chain-of-Thought (CoT) prompting scenarios (Appendix F).

*Table 3.* Ablation study on MIMIC-IV tasks comparing MedTPE, MedTPE without SSFT, and MedTPE without dependency-aware replacement (Rep.). Inference time is reported in minutes.

| Model | ICU Mortality | | Phenotyping | |
|---|---|---|---|---|
| | F1 ↑ | Time ↓ | F1 ↑ | Time ↓ |
| Llama3-1B | 0.033 | 66.0 | 0.050 | 50.2 |
| + MedTPE | **0.109** | **39.3** | **0.114** | **22.3** |
| + MedTPE w.o. SSFT | 0.000 | 56.8 | 0.007 | 41.6 |
| + MedTPE w.o. Rep. | 0.098 | 79.2 | 0.074 | 42.8 |
| Qwen2.5-1.5B | 0.122 | 38.3 | 0.201 | 29.7 |
| + MedTPE | **0.122** | **23.5** | **0.218** | **13.0** |
| + MedTPE w.o. SSFT | 0.031 | 55.5 | 0.064 | 60.0 |
| + MedTPE w.o. Rep. | 0.120 | 32.7 | 0.187 | 13.6 |

## 5.3. RQ2: Ablation Study

We evaluated MedTPE's core components via an ablation study in Table 3. Omitting Self-Supervised Fine-Tuning ("w.o. SSFT") causes catastrophic performance collapse. Without SSFT, average-initialised embeddings act as out-of-distribution noise, triggering hallucinations that negate compression efficiency. Furthermore, omitting dependency-aware replacement ("w.o. Rep.") by expanding the vocabulary and embedding matrices degrades accuracy and increases latency. Retaining original tokens introduces semantic ambiguity as they compete with new TPE tokens during generation, and unnecessarily expands structural matrices. Ultimately, both SSFT and strategic token replacement are indispensable to prevent generation confusion, minimise computational overhead, and preserve model capabilities.

## 5.4. RQ3: Evaluation of the Setup Cost

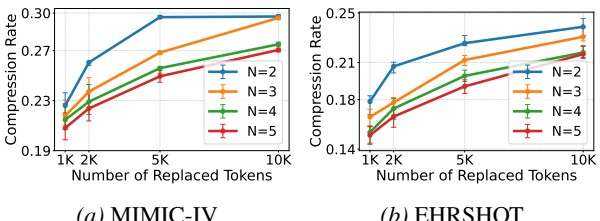

*(a) MIMIC-IV*     *(b) EHRSHOT*

*Figure 3.* Analysis of cost for MedTPE integration. Each curve shows the CR achieved with different numbers of replaced tokens and $N$-gram configurations ($N = 2, 3, 4, 5$), evaluated using the Qwen2.5 tokeniser on (a) MIMIC-IV and (b) EHRSHOT.

Figure 3 presents the CR achieved by MedTPE across various $N$-gram configurations and replacement budgets. We observe that the CR plateaus at approximately 30%, which likely represents the empirical bound for lossless compression on the MIMIC-IV and EHRSHOT datasets. This saturation aligns with Shannon's Source Coding Theorem (Shannon, 1948), which dictates that lossless reduction is strictly limited by the inherent entropy of the source text (Li, 2025). Our results demonstrate that a budget of 5,000 tokens offers the optimal trade-off for approaching this bound. Beyond this threshold, the marginal gain in compression diminishes significantly, as the remaining patterns fall into the 'long tail' of the clinical term distribution and yield diminishing gains in compression effectiveness (Portelli et al., 2022). Furthermore, to achieve the optimal gains, it is necessary to construct the vocabulary in the target domain (Appendix I).

In addition, the results indicate that bigrams ($N = 2$) consistently yield the highest compression rate across all settings. Although larger $N$-grams (e.g., "blood pressure monitor") are semantically richer, they are statistically rarer than their constituent bigrams ("blood pressure", "pressure monitor") (Clark et al., 2013). Given a fixed replacement budget ($M = 5,000$), allocating slots to high-frequency bigrams maximises the total number of merge operations across the corpus, whereas lower-frequency multi-word patterns offer diminishing returns. We find that a budget of 5,000 tokens achieves an optimal balance between computational overhead and compression performance. This addition represents only 3.3% of the Qwen-2.5 vocabulary and requires fine-tuning a negligible fraction of model parameters ($\approx 0.5$–1.0%). Supported by the qualitative analysis in Appendix G, our findings demonstrate that MedTPE achieves substantial efficiency gains by targeting high-frequency clinical units with minimal architectural modification. The cost of each setup step for MedTPE is shown in Appendix Table 7.

## 5.5. RQ4: Evaluation across Context-length

Figure 4 illustrates the predictive stability of MedTPE compared to original tokenisers across varying context scales.

*Table 4.* Generalisation evaluation of MedTPE on clinical narrative, scientific, and financial domains. Metrics include Accuracy (Acc.), F1, FCR, ROUGE (R-1, R-2, R-L), BERTScore (BS), and inference time.

*(a)* Evaluation of MedTPE on clinical narratives. Time is reported in minutes.

| Model | MDC Coding | | | | 30-day Readmission | | | |
|---|---|---|---|---|---|---|---|---|
| | F1 ↑ | FCR ↑ | Time ↓ | CR ↑ | F1 ↑ | FCR ↑ | Time ↓ | CR ↑ |
| Llama3-1B | $0.019_{\pm0.001}$ | 0.763 | 11.7 | – | $0.201_{\pm0.011}$ | 0.702 | 13.2 | – |
| + LLMLingua2 | $0.010_{\pm0.001}$ | 0.633 | 6.9 | 26.1% | $0.174_{\pm0.010}$ | 0.692 | 11.1 | 25.7% |
| + MedTPE | $0.048_{\pm0.003}$ | 0.994 | 2.7 | 26.1% | $0.302_{\pm0.010}$ | 0.916 | 9.0 | 25.7% |
| Qwen2.5-1.5B | $0.035_{\pm0.004}$ | 1.0 | 9.3 | – | $0.298_{\pm0.013}$ | 0.991 | 8.0 | – |
| + LLMLingua2 | $0.040_{\pm0.002}$ | 0.928 | 4.1 | 25.1% | $0.313_{\pm0.012}$ | 0.951 | 9.7 | 27.5% |
| + MedTPE | $0.036_{\pm0.003}$ | 0.999 | 3.5 | 25.1% | $0.300_{\pm0.008}$ | 1.000 | 5.1 | 27.5% |

*(b)* Evaluation of MedTPE across scientific (ARC-Challenge) and financial (ECTSum) domains. Time is reported in seconds.

| Model | ARC-Challenge | | | | ECTSum | | | | |
|---|---|---|---|---|---|---|---|---|---|
| | Acc.↑ | F1↑ | FCR↑ | Time↓ | R-1↑ | R-2↑ | R-L↑ | BS↑ | Time↓ |
| Llama3-1B | $0.473_{\pm0.015}$ | $0.474_{\pm0.015}$ | 0.992 | 21.2 | $0.110_{\pm0.004}$ | $0.035_{\pm0.002}$ | $0.070_{\pm0.002}$ | $0.818_{\pm0.001}$ | 113.2 |
| + LLMLingua2 | $0.165_{\pm0.011}$ | $0.181_{\pm0.010}$ | 0.846 | 88.0 | $0.123_{\pm0.003}$ | $0.032_{\pm0.001}$ | $0.071_{\pm0.002}$ | $0.718_{\pm0.001}$ | 165.6 |
| + MedTPE | $0.427_{\pm0.014}$ | $0.436_{\pm0.014}$ | 0.999 | 5.0 | $0.123_{\pm0.004}$ | $0.034_{\pm0.001}$ | $0.077_{\pm0.002}$ | $0.821_{\pm0.001}$ | 94.4 |
| Qwen2.5-1.5B | $0.635_{\pm0.014}$ | $0.643_{\pm0.014}$ | 0.999 | 9.3 | $0.140_{\pm0.004}$ | $0.038_{\pm0.002}$ | $0.092_{\pm0.003}$ | $0.824_{\pm0.001}$ | 107.9 |
| + LLMLingua2 | $0.259_{\pm0.013}$ | $0.230_{\pm0.019}$ | 0.982 | 16.7 | $0.127_{\pm0.004}$ | $0.032_{\pm0.002}$ | $0.082_{\pm0.002}$ | $0.819_{\pm0.001}$ | 216.7 |
| + MedTPE | $0.607_{\pm0.014}$ | $0.612_{\pm0.014}$ | 0.997 | 4.7 | $0.140_{\pm0.004}$ | $0.037_{\pm0.002}$ | $0.095_{\pm0.003}$ | $0.824_{\pm0.001}$ | 96.1 |

*(c)* Evaluation of MedTPE on the Chinese-language dataset (CMedQA2). Inference time is reported in seconds.

| Model | R-1↑ | R-2↑ | R-L↑ | BS↑ | Time↓ |
|---|---|---|---|---|---|
| Llama3-1B | $0.028_{\pm0.002}$ | $0.008_{\pm0.001}$ | $0.027_{\pm0.002}$ | $0.844_{\pm0.000}$ | 127.4 |
| + LLMLingua2 | $0.020_{\pm0.001}$ | $0.006_{\pm0.001}$ | $0.019_{\pm0.001}$ | $0.825_{\pm0.000}$ | 288.1 |
| + MedTPE | $0.027_{\pm0.002}$ | $0.007_{\pm0.001}$ | $0.026_{\pm0.002}$ | $0.844_{\pm0.000}$ | 113.3 |
| Qwen2.5-1.5B | $0.032_{\pm0.002}$ | $0.007_{\pm0.001}$ | $0.031_{\pm0.002}$ | $0.860_{\pm0.000}$ | 83.3 |
| + LLMLingua2 | $0.022_{\pm0.002}$ | $0.007_{\pm0.001}$ | $0.021_{\pm0.002}$ | $0.846_{\pm0.000}$ | 141.4 |
| + MedTPE | $0.030_{\pm0.002}$ | $0.007_{\pm0.001}$ | $0.029_{\pm0.002}$ | $0.860_{\pm0.000}$ | 67.3 |

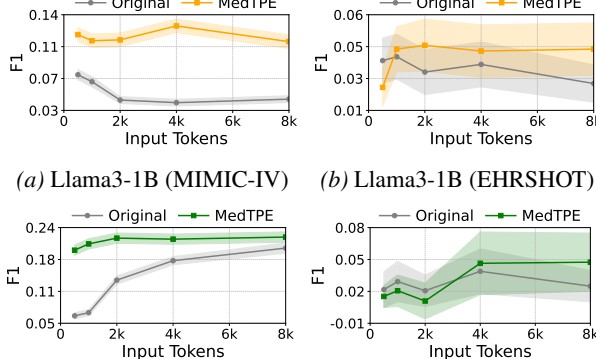

*(a)* Llama3-1B (MIMIC-IV)   *(b)* Llama3-1B (EHRSHOT)

*(c)* Qwen2.5-1.5B (MIMIC-IV) *(d)* Qwen2.5-1.5B (EHRSHOT)

*Figure 4.* Context length robustness of MedTPE. Each plot shows the mean F1 score with 95% confidence interval (shaded areas).

Evaluated on MIMIC-IV and EHRSHOT, MedTPE consistently maintains or surpasses baseline performance, substantiating its robustness across diverse sequence lengths. This reliability is further enhanced when integrated with test-time scaling strategies (Appendix K), allowing the effective leverage of computational budgets. Consequently, MedTPE enables predictive performance to scale alongside input length, offering a robust solution for clinical applications over highly frequent or longitudinal patient records.

## 5.6. RQ5: Evaluation of Generalisability

To evaluate the generalisability of MedTPE beyond structured EHR sequences, we assessed its performance across three distinct textual domains: clinical narratives, scientific reasoning, and financial summarisation. Within the clinical domain, we utilised the MIMIC-IV-Note dataset (Johnson et al., 2023) for two tasks: 30-day readmission prediction from discharge summaries and coding of the main diagnostic category (MDC), which requires mapping unstructured text to 25 standardised diagnostic groups. To test cross-domain robustness, we further added the ARC-Challenge dataset (Clark et al., 2018), comprising science questions that require logical reasoning, and the ECTSum dataset (Mukherjee et al., 2022), which requires the generation of concise summaries from lengthy financial earnings call transcripts. To evaluate cross-lingual transferability, we incorporated the Chinese Medical Question Answer Matching v2 (CMedQA2) dataset (Zhang et al., 2018). Following the evaluation protocol in (Jiang et al., 2024), we formulated this as a text generation task. We include LLMLingua2 (Pan et al., 2024) as the prompt compression baseline.

As shown in Table 4, MedTPE demonstrates generalisability, delivering substantial efficiency gains with minimal impact on predictive performance. On clinical narratives with typographical errors and artefacts, MedTPE maintains its effectiveness, confirming its robustness in processing noisy,

real-world documentation. While LLMLingua2 succeeds in this context by removing linguistic redundancies, MedTPE consistently achieves superior efficiency gains while maintaining comparable or higher predictive performance.

In the scientific domain, the method achieves high accuracy while reducing inference latency by 49.4% to 76.4%. Similarly, in the financial summarisation task, MedTPE consistently improves inference efficiency across diverse LLM families and parameter scales. In particular, for Llama-3-1B, MedTPE yields higher R-1 and R-L scores than the baseline, suggesting that the increased information density of TPE tokens may help smaller models to attend to salient concepts, even when applied to out-of-domain financial text.

MedTPE also exhibits cross-lingual transferability on the Chinese language CMedQA2. Unlike LLMLingua2, which suffers performance and latency degradation, MedTPE preserves baseline predictive performance while accelerating inference, enabling effective compression of non-English texts without language-specific retraining.

## 6. Conclusion

In this study, we proposed MedTPE, an efficient and lossless compression method that addresses the challenge of long EHR sequences by merging token pairs yielded from the general tokeniser into medical tokens. MedTPE achieves a substantial gain in inference efficiency without increasing the parameter size or sacrificing performance. Empirical evaluations across highly frequent and longitudinal EHR sequences and clinical narratives demonstrate that MedTPE effectively enhances the inference efficiency and robustness of pre-trained LLMs for clinical tasks.

While MedTPE provides efficiency gains, it relies on modifications to the tokeniser and embedding matrix. This process requires direct access to and alteration of the tokeniser, which may be impractical in closed-source models or production environments constrained by a fixed vocabulary. In addition, our qualitative analysis of Meditron3-8B under CoT prompting reveals instruction fragility in medically continual-pretrained models, which can be highly sensitive to input distribution shifts with CoT prompting. Integrating MedTPE with CoT prompt or advanced reasoning frameworks remains a current limitation of our methodology and represents a valuable direction for future exploration. Future work will explore instruction-aware token-pair selection and optimisation that preserves both clinical reasoning coherence and output-format compliance under CoT prompting.

## Acknowledgements

Mingcheng Zhu was supported by the Clarendon Fund Scholarship. Tingting Zhu was supported by the Royal Academy of Engineering under the Research Fellowship scheme.

## Impact Statement

The TPE vocabularies developed in this study are mined from specific, high-resource databases (e.g., MIMIC-IV, EHRSHOT) and inherently reflect localised clinical documentation practices. Consequently, deploying these tailored models in under-resourced settings or different geographic regions without rigorous re-validation introduces a distinct risk of dataset bias. If a model utilising these specific embeddings misinterprets divergent clinical terminologies or regional shorthand, it could inadvertently exacerbate existing healthcare disparities. Therefore, localised validation and vocabulary re-alignment are critical prerequisites for safe, real-world deployment.

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

## A. Algorithm of Dependency-aware Replacement

---

**Algorithm 1** Dependency-aware Replacement

---

**Input:** Original vocabulary $\mathcal{V}$, TPE candidates $\mathcal{V}_{\text{TPE}}$, budget $M$
**Output:** Optimised vocabulary $\mathcal{V}^\star$
**Step 1: Formulate Insertion Set ($\mathcal{I}$)**
**for** each $d_j \in \mathcal{V}_{\text{TPE}}$ **do**
    Calculate $\text{score}(d_j)$ via Eq. (9)
**end for**
$\mathcal{I} \leftarrow \text{Top-}M(\mathcal{V}_{\text{TPE}}, \text{score})$
**Step 2: Identify Dependencies ($\mathcal{D}$)**
Initialize $\mathcal{D} \leftarrow \emptyset$
**for** each $d_j \in \mathcal{I}$ **do**
    Retrieve merge path components $MP(d_j)$
    $\mathcal{D} \leftarrow \mathcal{D} \cup \{x \mid x \in MP(d_j)\}$ {Preserve recursive dependencies (Eq. (10))}
**end for**
**Step 3: Select Eviction Set ($\mathcal{E}$)**
$\mathcal{U} \leftarrow \mathcal{V} \setminus \mathcal{D}$ {Identify unprotected tokens}
$\mathcal{S} \leftarrow \text{SortAscending}(\mathcal{U}, \text{freq})$
$\mathcal{E} \leftarrow \mathcal{S}[1 : M]$ {Select least frequent candidates}
**Step 4: Construct Final Vocabulary ($\mathcal{V}^\star$)**
$\mathcal{V}^\star \leftarrow (\mathcal{V} \setminus \mathcal{E}) \cup \mathcal{I}$ {Merge vocabularies (Eq. (11))}
**return** $\mathcal{V}^\star$

---

## B. Algorithm of MedTPE Integration

## C. Implementation Details

*Table 5.* Statistics of Datasets

| Statistics | MIMIC-IV | EHRSHOT |
|---|---|---|
| # Patients | 42,117 | 6,739 |
| # Visits | 57,523 | 921,499 |
| # Distinct Events | 42,200 | 31,252 |
| # Average Tokens (Qwen2) | 13,456 | 190,660 |
| # Average Tokens (Llama3) | 12,145 | 158,004 |
| # Average Tokens (Gemma2) | 13,486 | 186,604 |

### C.1. Datasets

For clinical datasets, we transform raw EHR data into structured event streams following the Medical Event Data Standard (MEDS) (Arnrich et al., 2024), capturing each patient's record as a chronologically ordered sequence of timestamped clinical events. Following previous works (Mesinovic et al., 2026; Chen et al., 2026), both datasets are split at the patient level to ensure that records from the same patient appear in a single subset, guaranteeing the independence between training, validation, and test sets. Patient IDs are matched across datasets according to

*Table 6.* Summary of dataset splits for all tasks.

| Task (Dataset) | Domain | Size | | | # Labels |
|---|---|---|---|---|---|
| | | **Train** | **Validation** | **Test** | |
| ICU Mortality (MIMIC-IV) | Clinical (EHR) | 46,106 | 5,602 | 5,815 | 2 |
| Phenotyping (MIMIC-IV) | Clinical (EHR) | 26,937 | 3,267 | 3,420 | 25 |
| 30-day Readmission (EHRSHOT) | Clinical (EHR) | 2,608 | 2,206 | 2,189 | 2 |
| 1-year Pancreatic Cancer (EHRSHOT) | Clinical (EHR) | 2,576 | 2,215 | 2,220 | 2 |
| 30-day Readmission (MIMIC-IV-Note) | Clinical (Narrative) | 25,670 | 3,166 | 3,180 | 2 |
| MDC Classification (MIMIC-IV-Note) | Clinical (Narrative) | 17,935 | 2,280 | 2,258 | 25 |
| Chinese Medical QA Matching v2 (cMedQA2) | Clinical (Chinese) | 100,000 | 4,000 | 4,000 | Free-text |
| ARC-Challenge (ARC) | Scientific | 1,119 | 299 | 1,172 | 4 |
| ECT Summarisation (ECTSUM) | Financial | 1,681 | 249 | 495 | Free-text |

---

**Algorithm 2** MedTPE Integration with LLM

---

**Input:** Medical corpus $\mathcal{C}$, original tokeniser $\tau$, LLM $F = (\tau, \Theta)$, budget $M$, input $s$
**Output:** LLM with MedTPE $F^\star = (\tau^\star, \Theta^\star)$
**Step 1: Build MedTPE Tokeniser**
Extract original vocab $\mathcal{V}$ and merge table $M_{\text{orig}}$ from $\tau$
Score all $N$-grams ($2 \leq N \leq N_{\max}$) via Eq. 9
$\mathcal{I} \leftarrow$ top-$M$ TPE tokens
$\mathcal{D} \leftarrow$ dependent set via Eq. 10
$\mathcal{E} \leftarrow M$ least-frequent tokens in $\mathcal{V} \setminus \mathcal{D}$
$\mathcal{V}^\star \leftarrow (\mathcal{V} \setminus \mathcal{E}) \cup \mathcal{I}$ via Eq. 11
$\mathcal{M}_{\text{TPE}} \leftarrow [\|_{d \in \mathcal{I}} MP(d)]$
Define $\tau^\star = (\tau, \mathcal{V}^\star, \mathcal{M}_{\text{TPE}})$
**Step 2: MedTPE Encoding**
$\mathcal{X} \leftarrow \tau(s)$
$\mathcal{X}^\star \leftarrow []$
$i \leftarrow 1$
**while** $i \leq |\mathcal{X}|$ **do**
  $j \leftarrow \max\{k \geq i : MP(x_i \ldots x_k) \subseteq \mathcal{M}_{\text{TPE}}\}$
  Append $(x_i \oplus \cdots \oplus x_j)$ to $\mathcal{X}^\star$
  $i \leftarrow j + 1$
**end while**
**Step 3: Self-supervised Fine-tuning**
Initialise embedding $e_d$ for $d \in \mathcal{I}$ with Eq. 12 and Eq. 13
Freeze $\Theta \setminus \{\mathbf{e}_d\}_{d \in \mathcal{I}}$
**for** each mini-batch $b \subset \mathcal{C}$ **do**
  $X^\star \leftarrow$ MedTPE-Tokenise($b$)
  $y_{\text{pseu}} \leftarrow \arg\max(\mathcal{F}(X^\star))$ {Generate pseudo-labels}
  Update $\{\mathbf{e}_d\}$ using cross-entropy loss with $y_{\text{pseu}}$
**end for**
**return** $F^\star = (\tau^\star, \Theta^\star)$

---

MEDS specifications to maintain consistency and ensure reproducibility (Arnrich et al., 2024). For MIMIC-IV, we use an 8:1:1 split for training, validation, and test sets, while for EHRSHOT, we use a 1:1:1 split.

For clinical narratives, we utilised the MIMIC-IV-Note dataset (Johnson et al., 2023), which provides a comprehensive collection of de-identified clinical notes associated with hospital admissions. Unlike the structured events in the core MIMIC-IV database, this dataset consists of unstructured text, including discharge summaries, radiology reports, and nursing notes, which often contain typographical errors and idiosyncratic human artefacts. We use the discharge summaries for our evaluation.

To evaluate the cross-domain generalisation of MedTPE beyond the clinical sphere, we extended our evaluation to the scientific and financial sectors using the ARC-Challenge and ECTSUM datasets. The ARC-Challenge (Clark et al., 2018) comprises grade-school science questions requiring complex reasoning, while ECTSUM (Mukherjee et al., 2022) focuses on the financial domain, requiring long-form summarisation of verbose earnings call transcripts to extract salient corporate facts. These datasets provide a diverse testing ground to verify that MedTPE's tokenisation logic is robust across varied linguistic structures. Table 6 summarises the training, validation, and test splits for all datasets, all of which were processed in full compliance with their respective licensing conditions.

### C.2. Training Details

We performed SSFT of all LLMs using a single NVIDIA A800 80G GPU, while inference experiments were conducted on a single NVIDIA A5500 24G GPU.

We fine-tuned the LLM using self-supervised learning, setting a fixed learning rate of $5 \times 10^{-5}$ with the AdamW optimiser. Training was performed with a batch size of 2 and gradient accumulation in 2 steps, producing an effective batch size of 4. The input and output sequence lengths were capped at 4,096 tokens. We did not conduct hyperparameter tuning. Instead, we adopted a configuration feasible for the largest model (Llama3-8B) on a single 80 GB GPU and applied it consistently across all LLMs. The learning rate schedule involved a linear warmup during the first 10% of the total training steps, followed by a cosine decay until the end of training. Model performance was validated every 1,000 steps, with early stopping applied if validation performance did not improve for 3 consecutive checks. To ensure

reproducibility, the random seed was set to 1. Only the embeddings of new tokens were trainable during fine-tuning, with all other embeddings and LLM layers frozen. The padding token was set to be identical to the EOS token. We did not perform hyperparameter tuning since the selected settings were constrained by hardware limits (e.g., batch size and sequence length), and no baselines required training, so a single plausible configuration was used throughout.

### C.3. Evaluation Details

For all tasks, the maximum input length was set to 8,192, and the maximum output length was set to 4,096 tokens. The amount of input information provided to each model was determined by the maximum input length allowed by the model's original tokeniser. The sampling temperature for LLM inference was set at 0.7, following common practice to balance generation diversity and output reliability for general-purpose LLMs (Du et al., 2025). All reported F1 scores were obtained by bootstrapping the test set 1,000 times to ensure a robust and reliable evaluation. We implement a dedicated embedding split module (Appendix D) to ensure that gradients are computed exclusively for the new embeddings, avoiding unnecessary memory usage and computation.

### C.4. Evaluated LLMs

We evaluated MedTPE across several LLM families to ensure broad architectural compatibility. Specifically, we used the Qwen2.5 (1.5B and 7B) models (Team, 2024), which were trained on an 18-trillion-token corpus with dual-stage supervised and RLHF fine-tuning. We also included Qwen3 (1.7B) model (Yang et al., 2025) into the evaluation. From the Llama3 family (Grattafiori et al., 2024), we selected the 1B and 8B variants. We also included Meditron3-8B (Sallinen et al., 2025), an open clinical LLM suite developed through continued pre-training of Llama3 in medical corpora for enhanced clinical decision support. All these families use Byte Pair Encoding (BPE) tokenisation (Chizhov et al., 2024). Finally, to evaluate MedTPE's versatility with different segmentation methods, we included Gemma2-2B (Team et al., 2024), which employs the SentencePiece tokeniser (Kudo & Richardson, 2018) for text processing.

### C.5. Setup Cost of MedTPE

Table 7 delineates the computational overhead associated with the MedTPE setup pipeline, comprising the Encoding, Replacement, and SSFT phases. The initial encoding and replacement steps are highly efficient, executing in a few minutes. The primary computational investment lies in the SSFT phase. However, because this phase freezes the core LLM backbone and updates only the newly introduced TPE embeddings (constituting roughly 0.5% to 1.0% of total parameters), it is efficient and remains a strictly one-time offline procedure. Overall training times scale with model capacity: processing 1B to 1.5B parameter models requires approximately 2 to 6 hours on a single NVIDIA RTX 6000 Ada GPU, while the larger 7B to 8B architectures require between 13 and 16.5 hours on a single NVIDIA A100 GPU.

### C.6. Code Availability

All the source code required to perform the experiments is available in the GitHub repository[1].

## D. Embedding Split Module

To enable supervised fine-tuning of only a subset of embedding vectors, we introduce the embedding split module. Fine-tuning a subset of embeddings within a single unified embedding matrix can result in unnecessary gradient computation for the frozen embeddings, leading to increased memory usage and computational overhead. Our module addresses this by explicitly partitioning the embedding set into two disjoint subsets:

$$\mathbf{E} = \mathbf{E}_{\text{orig, fixed}} \cup \mathbf{E}_{\text{TPE, trainable}},$$

where $\mathbf{E}_{\text{orig, fixed}}$ denotes the pre-trained token embeddings, which remain fixed, and $\mathbf{E}_{\text{TPE, trainable}}$ is the trainable embeddings of the new TPE tokens.

During the forward pass, the embeddings are retrieved by separately indexing both subsets and concatenating the results to form the complete embedding set $\mathbf{E}$. During the backward pass, gradient-based updates are restricted to the trainable subset $\mathcal{E}_{\text{TPE, trainable}}$, substantially improving memory and computational efficiency.

## E. Evaluation of MedTPE on More LLMs

To evaluate the versatility and robustness of MedTPE, we extended our evaluation to a broader range of LLMs across different model families, parameter scales (1B to 8B), and training domains. As detailed in the Table 8 (a)(b), MedTPE consistently enhances inference efficiency across both general-purpose models (Llama-3, Qwen-2.5, Qwen-3, Gemma-2) and domain-specific architectures (Meditron-3). Across all configurations, MedTPE achieves a substantial reduction in inference latency, ranging from 30.9% to 62.5%, regardless of the underlying tokenisation method or parameter count. Crucially, the predictive performance remains remarkably stable. Even as model size increases to 7B and 8B parameters, MedTPE preserves the high reliability of the original models, maintaining FCR values near or at 1.0. These results demonstrate that MedTPE is a highly adaptable and reliable framework for clinical LLM acceleration, offering

[1] https://github.com/JasonZuu/MedTPE

*Table 7.* Time breakdown for MedTPE processing phases on MIMIC-IV datasets.

| Model | Task | Encoding (min) | Replacement (min) | SSFT (h) |
|-------|------|----------------|-------------------|----------|
| Llama3-1B | ICU Mortality | 2.9 | 2.0 | 2.6 |
|  | ICU Phenotyping | 1.9 | 1.2 | 2.0 |
| Qwen2.5-1.5B | ICU Mortality | 2.3 | 0.2 | 4.3 |
|  | ICU Phenotyping | 1.4 | 0.4 | 5.7 |
| Qwen2.5-7B | ICU Mortality | 2.3 | 0.2 | 14.5 |
|  | ICU Phenotyping | 1.4 | 0.4 | 13.2 |
| Llama3-8B | ICU Mortality | 2.9 | 2.0 | 16.2 |
|  | ICU Phenotyping | 1.9 | 1.2 | 14.8 |
| Meditron3-8B | ICU Mortality | 2.9 | 2.0 | 13.3 |
|  | ICU Phenotyping | 1.9 | 1.2 | 14.8 |

consistent efficiency gains without compromising diagnostic precision.

To evaluate the scalability of MedTPE, we extended our experiments to larger architectural scales using the Qwen2.5-14B and 32B models. For these evaluations, we specifically selected the phenotyping prediction task, which is the most challenging benchmark in our study. As summarised in Table 8(c), MedTPE maintains its efficiency and efficacy as model capacity increases. Specifically, we observed substantial reductions in inference latency 36.3% for the 14B model and 40.9% for the 32B model—while preserving F1 scores with marginal degradation. These findings confirm that MedTPE's compression mechanism is highly compatible with large-scale architectures and remains robust even under the most demanding clinical reasoning requirements.

## F. Impact of CoT Prompting

We further evaluated the effectiveness of MedTPE under CoT prompting, as shown in Table 9. In all models and tasks, MedTPE consistently reduced inference time and achieved substantial sequence compression, with compression rates ranging from 22.8% to 32.4%. For LLMs larger than 7B parameters, F1 scores and format compliance rates were largely maintained with only marginal decreases. In contrast, smaller models such as Llama3-1B and Qwen2.5-1.5B exhibited notable improvements in both predictive performance and compliance with the output format when equipped with MedTPE. For example, Llama3-1B with CoT prompting in ICU mortality saw its F1 score increase from 0.030 to 0.122 and FCR from 0.231 to 0.999, while the inference time was reduced by more than 66%. Similar trends were observed across the EHRSHOT tasks and for Qwen2.5-1.5B, which demonstrated improvements in both efficiency and predictive performance.

An exception to the general CoT results is observed for Meditron3-8B on the MIMIC-IV phenotyping task. In this setting, MedTPE reduces the input length by 32.4%, but

the F1 score decreases from 0.176 to 0.137, the format compliance rate decreases from 0.987 to 0.852, and the inference time increases from 94.0 to 135.8 minutes. To better understand this degradation, we qualitatively inspected representative generated responses from Meditron3-8B with CoT prompting on ICU phenotyping. The analysis reveals one expected behaviour and two recurring failure modes, summarised in Table 10. Failure Mode A indicates reasoning hallucination: the model preserves the required output format, but the CoT rationale becomes misaligned with the final prediction. Failure Mode B indicates a loss of instruction adherence: the model continues generating clinical explanations but fails to produce the required JSON output. These behaviours are consistent with instruction fragility in medically continual-pretrained models, where changes to the input token distribution can disrupt either the reasoning trajectory or the formatting constraint. Therefore, although MedTPE is generally effective under CoT prompting, integrating token-pair compression with advanced reasoning prompts should be validated for each fine-tuned model.

## G. Analysis of TPE Tokens

To identify the primary drivers of the observed efficiency gains, we analysed the most frequent TPE tokens within the MIMIC-IV dataset. As illustrated in Table 11, the top-10 tokens are dominated by clinical units, physiological measurements, and frequent medical terminology. Notably, while the specific rankings differ slightly, Llama3 and Qwen2.5 produced identical top-10 token sets, primarily capturing vital sign descriptors (e.g., "mmHg", "Heart rate") and nursing observations (e.g., "Nonin", "Infusion"). Gemma2 similarly prioritises high-frequency substrings such as "Infusion" and various volumetric units.

This distribution indicates that MedTPE effectively identifies and merges semantically meaningful substrings that recur throughout clinical documentation. By representing these common patterns, such as blood pressure metrics and

*Table 8.* Assessment of LLMs with MedTPE. Mean and standard deviation of F1 scores are reported, calculated by bootstrapping the test set 1,000 times. Inference time is reported in minutes, with the percentage change shown relative to the original LLM. Llama3 series and Meditron3 use the same tokeniser, resulting in identical CR with prompt compression.

*(a)* Evaluation of LLMs with MedTPE on MIMIC-IV

| Model | ICU Mortality | | | | Phenotyping | | | |
|---|---|---|---|---|---|---|---|---|
| | $F1_{\pm std}\uparrow$ | FCR↑ | Time (Δ)↓ | CR↑ | $F1_{\pm std}\uparrow$ | FCR↑ | Time (Δ)↓ | CR↑ |
| Llama3-1B | $0.033_{\pm0.003}$ | 0.330 | 66.0 | - | $0.050_{\pm0.002}$ | 0.492 | 50.2 | - |
| + LLMLingua2 | $0.035_{\pm0.003}$ | 0.328 | 82.5 (25%) | 32.1% | $0.058_{\pm0.003}$ | 0.500 | 38.7(-22.9%) | 32.4% |
| + MedTPE | $0.109_{\pm0.006}$ | 0.891 | 39.3 (-40.5%) | 32.1% | $0.114_{\pm0.004}$ | 0.870 | 22.3 (-55.6%) | 32.4% |
| Qwen2.5-1.5B | $0.122_{\pm0.006}$ | 0.998 | 38.3 | - | $0.201_{\pm0.005}$ | 0.969 | 29.7 | - |
| + LLMLingua2 | $0.103_{\pm0.006}$ | 0.899 | 57.5 (50.1%) | 27.2% | $0.020_{\pm0.001}$ | 0.399 | 41.0 (38.0%) | 29.7% |
| + MedTPE | $0.122_{\pm0.006}$ | 1.000 | 23.5 (-38.6%) | 27.2% | $0.218_{\pm0.005}$ | 0.999 | 13.0 (-56.2%) | 29.7% |
| Qwen3-1.7B | $0.130_{\pm0.006}$ | 1.000 | 66.0 | - | $0.139_{\pm0.004}$ | 0.998 | 43.8 | - |
| + LLMLingua2 | $0.127_{\pm0.008}$ | 0.983 | 138.7 (110.2%) | 25.4% | $0.074_{\pm0.003}$ | 0.964 | 145.6 (232.4%) | 29.7% |
| + MedTPE | $0.128_{\pm0.006}$ | 1.000 | 43.7 (-33.8%) | 25.4% | $0.137_{\pm0.003}$ | 0.987 | 23.4 (-46.6%) | 29.7% |
| Gemma2-2B | $0.003_{\pm0.002}$ | 0.592 | 81.0 | - | $0.132_{\pm0.003}$ | 0.994 | 32.9 | - |
| + LLMLingua2 | $0.003_{\pm0.001}$ | 0.216 | 101.6 (25.4%) | 23.7% | $0.023_{\pm0.001}$ | 0.478 | 69.0 (109.7%) | 24.0% |
| + MedTPE | $0.014_{\pm0.004}$ | 0.851 | 56.0 (-30.9%) | 23.7% | $0.119_{\pm0.003}$ | 0.987 | 16.5 (-49.8%) | 24.0% |
| Qwen2.5-7B | $0.137_{\pm0.007}$ | 0.994 | 308.1 | - | $0.189_{\pm0.004}$ | 0.995 | 165.3 | - |
| + LLMLingua2 | $0.132_{\pm0.007}$ | 0.997 | 266.5 (-13.5%) | 27.2% | $0.082_{\pm0.002}$ | 0.841 | 151.7 (-8.2%) | 29.7% |
| + MedTPE | $0.132_{\pm0.006}$ | 0.990 | 183.1 (-40.6%) | 27.2% | $0.174_{\pm0.005}$ | 0.988 | 94.8 (-42.6%) | 29.7% |
| Llama3-8B | $0.123_{\pm0.007}$ | 0.994 | 378.4 | - | $0.173_{\pm0.004}$ | 0.999 | 193.8 | - |
| + LLMLingua2 | $0.123_{\pm0.006}$ | 0.968 | 389.8 (3.0%) | 32.1% | $0.065_{\pm0.002}$ | 0.780 | 231.1 (19.2%) | 32.4% |
| + MedTPE | $0.124_{\pm0.007}$ | 0.994 | 181.2 (-52.1%) | 32.1% | $0.164_{\pm0.004}$ | 0.999 | 95.7 (-50.6%) | 32.4% |
| Meditron3-8B | $0.114_{\pm0.009}$ | 0.996 | 152.6 | - | $0.183_{\pm0.004}$ | 0.998 | 84.4 | - |
| + LLMLingua2 | $0.063_{\pm0.006}$ | 0.763 | 149.6 (-2.0%) | 32.1% | $0.046_{\pm0.002}$ | 0.508 | 125.4 (48.6%) | 32.4% |
| + MedTPE | $0.115_{\pm0.009}$ | 0.997 | 96.0 (-37.1%) | 32.1% | $0.204_{\pm0.004}$ | 0.999 | 52.6 (-37.7%) | 32.4% |

*(b)* Evaluation of LLMs with MedTPE on EHRSHOT

| Model | 30-day Readmission | | | | 1-year Pancreatic Cancer | | | |
|---|---|---|---|---|---|---|---|---|
| | $F1_{\pm std}\uparrow$ | FCR↑ | Time (Δ)↓ | CR↑ | $F1_{\pm std}\uparrow$ | FCR↑ | Time (Δ)↓ | CR↑ |
| Llama3-1B | $0.098_{\pm0.008}$ | 0.425 | 28.1 | - | $0.039_{\pm0.006}$ | 0.748 | 28.5 | - |
| + LLMLingua2 | $0.071_{\pm0.008}$ | 0.326 | 57.8(105.7%) | 27.5% | $0.024_{\pm0.005}$ | 0.503 | 39.3 (37.9%) | 28.3% |
| + MedTPE | $0.191_{\pm0.011}$ | 0.877 | 16.6 (-40.7%) | 27.5% | $0.047_{\pm0.006}$ | 0.900 | 15.9 (-44.3%) | 28.3% |
| Qwen2.5-1.5B | $0.205_{\pm0.011}$ | 0.989 | 15.2 | - | $0.047_{\pm0.010}$ | 0.877 | 30.8 | - |
| + LLMLingua2 | $0.176_{\pm0.012}$ | 0.860 | 25.3 (66.4%) | 22.8% | $0.044_{\pm0.007}$ | 0.860 | 32.1 (4.2%) | 23.1% |
| + MedTPE | $0.209_{\pm0.011}$ | 0.990 | 10.0 (-33.9%) | 22.8% | $0.066_{\pm0.017}$ | 0.947 | 11.5 (-62.5%) | 23.1% |
| Gemma2-2B | $0.190_{\pm0.011}$ | 0.905 | 19.4 | - | $0.089_{\pm0.013}$ | 0.983 | 19.0 | - |
| + LLMLingua2 | $0.036_{\pm0.006}$ | 0.207 | 26.3 (35.6%) | 20.5% | $0.045_{\pm0.006}$ | 0.854 | 21.3 (4.2%) | 21.2% |
| + MedTPE | $0.206_{\pm0.012}$ | 0.973 | 10.9 (-43.8%) | 20.5% | $0.064_{\pm0.009}$ | 0.950 | 14.6 (-23.2%) | 21.2% |
| Qwen2.5-7B | $0.211_{\pm0.011}$ | 1.000 | 95.7 | - | $0.126_{\pm0.033}$ | 1.000 | 78.2 | - |
| + LLMLingua2 | $0.212_{\pm0.009}$ | 0.990 | 95.7 (0.0%) | 22.8% | $0.136_{\pm0.030}$ | 0.994 | 92.8 (18.7%) | 23.1% |
| + MedTPE | $0.212_{\pm0.011}$ | 0.996 | 59.1 (-38.3%) | 22.8% | $0.132_{\pm0.030}$ | 0.996 | 54.9 (-29.8%) | 23.1% |
| Llama3-8B | $0.211_{\pm0.011}$ | 0.998 | 126.1 | - | $0.055_{\pm0.008}$ | 0.999 | 152.7 | - |
| + LLMLingua2 | $0.198_{\pm0.011}$ | 0.945 | 182.5 (44.7%) | 27.5% | $0.061_{\pm0.009}$ | 0.978 | 190.4 (24.7%) | 28.3% |
| + MedTPE | $0.211_{\pm0.011}$ | 0.995 | 63.2 (-49.9%) | 27.5% | $0.068_{\pm0.009}$ | 0.989 | 77.6 (-49.2%) | 28.3% |
| Meditron3-8B | $0.212_{\pm0.012}$ | 0.995 | 57.4 | - | $0.055_{\pm0.009}$ | 0.989 | 59.4 | - |
| + LLMLingua2 | $0.118_{\pm0.010}$ | 0.587 | 93.4 (62.7%) | 27.5% | $0.042_{\pm0.007}$ | 0.825 | 77.3 (30.1%) | 28.3% |
| + MedTPE | $0.211_{\pm0.011}$ | 1.000 | 36.7 (-36.1%) | 27.5% | $0.040_{\pm0.007}$ | 0.989 | 37.6 (-36.7%) | 28.3% |

*(c)* Scalability evaluation of MedTPE on the phenotyping task (MIMIC-IV).

| Model | $F1_{\pm std}\uparrow$ | FCR ↑ | Time (Δ) ↓ | CR ↑ |
|---|---|---|---|---|
| Qwen2.5-14B | $0.177_{\pm0.004}$ | 1.0 | 306.6 | – |
| + MedTPE | $0.172_{\pm0.005}$ | 1.0 | 168.2 (-36.3%) | 27.2% |
| Qwen2.5-32B* | $0.205_{\pm0.004}$ | 1.0 | 45.2 | – |
| + MedTPE | $0.191_{\pm0.004}$ | 0.998 | 26.7 (-40.9%) | 27.2% |

[*]Evaluated using eight A100 GPUs.

infusion statuses, MedTPE achieves substantial sequence compression. This targeted reduction in input length allows

*Table 9.* Assessment of LLMs with MedTPE using CoT prompting. Mean and standard deviation of F1 scores are reported, calculated by bootstrapping the test set 1,000 times. Inference time is reported in minutes, with the percentage change shown relative to the original LLM. Llama3 series and Meditron3 use the same tokeniser, resulting in identical CR with prompt compression.

*(a)* Evaluation of LLMs with MedTPE on MIMIC-IV with CoT Prompting

| LLM with CoT | ICU Mortality | | | | Phenotyping | | | |
|---|---|---|---|---|---|---|---|---|
| | $F1_{\pm std}\uparrow$ | FCR$\uparrow$ | Time ($\Delta$)$\downarrow$ | CR$\uparrow$ | $F1_{\pm std}\uparrow$ | FCR$\uparrow$ | Time ($\Delta$)$\downarrow$ | CR$\uparrow$ |
| Llama3-1B | $0.030_{\pm 0.003}$ | 0.231 | 69.8 | - | $0.050_{\pm 0.002}$ | 0.458 | 49.7 | - |
| + MedTPE | $0.122_{\pm 0.006}$ | 0.999 | 23.5 (-66.3%) | 32.1% | $0.104_{\pm 0.004}$ | 0.832 | 24.0 (-51.7%) | 32.4% |
| Qwen2.5-1.5B | $0.033_{\pm 0.004}$ | 0.441 | 206.0 | - | $0.027_{\pm 0.002}$ | 0.388 | 130.5 | - |
| + MedTPE | $0.122_{\pm 0.006}$ | 0.999 | 23.5 (-88.6%) | 27.2% | $0.135_{\pm 0.004}$ | 0.787 | 38.0 (-70.9%) | 29.7% |
| Gemma2-2B | $0.001_{\pm 0.001}$ | 0.523 | 105.7 | - | $0.017_{\pm 0.001}$ | 0.498 | 58.3 | - |
| + MedTPE | $0.012_{\pm 0.003}$ | 0.732 | 83.5 (-21.0%) | 23.7% | $0.050_{\pm 0.002}$ | 0.893 | 19.6 (-66.4%) | 24.0% |
| Qwen2.5-7B | $0.131_{\pm 0.006}$ | 0.991 | 403.6 | - | $0.189_{\pm 0.004}$ | 0.980 | 261.6 | - |
| + MedTPE | $0.129_{\pm 0.006}$ | 0.979 | 222.9 (-44.8%) | 27.2% | $0.172_{\pm 0.004}$ | 0.979 | 113.0 (-56.8%) | 29.7% |
| Llama3-8B | $0.135_{\pm 0.007}$ | 0.990 | 522.1 | - | $0.176_{\pm 0.004}$ | 0.987 | 285.9 | - |
| + MedTPE | $0.129_{\pm 0.006}$ | 0.996 | 195.9 (-62.5%) | 32.1% | $0.175_{\pm 0.004}$ | 0.997 | 106.0 (-62.9%) | 32.4% |
| Meditron3-8B | $0.106_{\pm 0.008}$ | 0.938 | 190.1 | - | $0.176_{\pm 0.004}$ | 0.987 | 94.0 | - |
| + MedTPE | $0.103_{\pm 0.008}$ | 0.880 | 126.0 (-33.7%) | 32.1% | $0.137_{\pm 0.004}$ | 0.852 | 135.8 (44.6%) | 32.4% |

*(b)* Evaluation of LLMs with MedTPE on EHRSHOT with CoT Prompting

| LLM with CoT | 30-day Readmission | | | | 1-year Pancreatic Cancer | | | |
|---|---|---|---|---|---|---|---|---|
| | $F1_{\pm std}\uparrow$ | FCR$\uparrow$ | Time ($\Delta$)$\downarrow$ | CR$\uparrow$ | $F1_{\pm std}\uparrow$ | FCR$\uparrow$ | Time ($\Delta$)$\downarrow$ | CR$\uparrow$ |
| Llama3-1B | $0.067_{\pm 0.007}$ | 0.297 | 27.9 | - | $0.033_{\pm 0.006}$ | 0.630 | 29.8 | - |
| + MedTPE | $0.183_{\pm 0.011}$ | 0.862 | 17.0 (-39.0%) | 27.5% | $0.044_{\pm 0.006}$ | 0.850 | 18.2 (-38.8%) | 28.3% |
| Qwen2.5-1.5B | $0.069_{\pm 0.008}$ | 0.396 | 80.1 | - | $0.032_{\pm 0.006}$ | 0.746 | 52.2 | - |
| + MedTPE | $0.211_{\pm 0.011}$ | 0.986 | 10.4 (-87.1%) | 22.8% | $0.027_{\pm 0.008}$ | 0.841 | 15.9 (-69.5%) | 23.1% |
| Gemma2-2B | $0.020_{\pm 0.005}$ | 0.423 | 25.7 | - | $0.031_{\pm 0.008}$ | 0.708 | 24.0 | - |
| + MedTPE | $0.181_{\pm 0.013}$ | 0.889 | 19.1 (-25.7%) | 20.5% | $0.041_{\pm 0.009}$ | 0.868 | 16.5 (-31.3%) | 21.2% |
| Qwen2.5-7B | $0.212_{\pm 0.011}$ | 0.995 | 127.4 | - | $0.091_{\pm 0.020}$ | 0.995 | 125.9 | - |
| + MedTPE | $0.211_{\pm 0.011}$ | 0.992 | 65.7 (-48.4%) | 22.8% | $0.109_{\pm 0.025}$ | 0.992 | 65.7 (-47.8%) | 23.1% |
| Llama3-8B | $0.213_{\pm 0.011}$ | 1.0 | 159.4 | - | $0.058_{\pm 0.009}$ | 0.998 | 199.7 | - |
| + MedTPE | $0.209_{\pm 0.011}$ | 0.990 | 72.6 (-55.4%) | 27.5% | $0.060_{\pm 0.009}$ | 0.984 | 95.7 (-52.1%) | 28.3% |
| Meditron3-8B | $0.176_{\pm 0.011}$ | 0.907 | 73.6 | - | $0.040_{\pm 0.007}$ | 0.850 | 84.7 | - |
| + MedTPE | $0.207_{\pm 0.011}$ | 0.979 | 42.9 (-41.8%) | 27.5% | $0.044_{\pm 0.008}$ | 0.975 | 49.5 (-41.5%) | 28.3% |

the models to retain essential diagnostic information while significantly decreasing the computational cost of processing lengthy clinical records.

# H. Evaluating Baselines with Embedding Fine-Tuning

To ensure a strictly equitable comparison, we conducted an experiment applying the embedding fine-tuning (Emb-FT) step to the prompt compression baselines. While MedTPE inherently utilises SSFT to align new tokens, methods like LLMLingua2 and T5Summary do not traditionally undergo domain-specific embedding adaptation. To isolate the impact of this tuning phase, we augmented both baselines with an equivalent Emb-FT step on the target domain, evaluating them on the MIMIC-IV Phenotyping task using the Llama3-1B model. ZeTT was explicitly excluded from this setup, as its architecture relies on a hypernetwork to generate dynamic embeddings rather than tuning static representations.

As shown in Table 12, introducing this fine-tuning step to the baselines fails to close the performance gap. Emb-FT yields a marginal F1 improvement for T5Summary (from

0.078 to 0.081) and degrades LLMLingua2's predictive performance (from 0.058 to 0.017). This degradation likely occurs because traditional prompt compression methods aggressively prune tokens, resulting in discontinuous text. Fine-tuning representations on this fragmented context disrupts the LLM's pre-trained semantic space. Conversely, MedTPE preserves semantic continuity and significantly outperforms these augmented baselines, confirming that its efficacy stems from its dependency-aware token construction rather than the fine-tuning step alone.

# I. Cross-Domain Transferability of MedTPE

To evaluate MedTPE's sensitivity to the corpus used for vocabulary mining and representation alignment, we conducted a cross-domain transferability experiment. Specifically, we directly applied the MedTPE tokeniser and embeddings optimised exclusively on the MIMIC-IV dataset to the EHRSHOT evaluation suites without any adaptation.

Results in Table 13 reveal a limitation regarding the direct cross-domain transferability of MedTPE. Applying the MIMIC-IV-derived MedTPE model to the EHRSHOT

*Table 10.* Qualitative failure-mode analysis of Meditron3-8B with MedTPE under CoT prompting on the MIMIC-IV phenotyping task. Representative outputs are shortened for readability while preserving the relevant reasoning and output-format generation.

| Category | Representative behaviour | Interpretation |
|---|---|---|
| Correct Prediction | *"The patient's EHR data suggest that they experienced respiratory failure, tachycardia, and hypotension, which are consistent with acute myocardial infarction... Laboratory results showed elevated troponin levels, which are consistent with cardiac damage. Based on this information, the predicted phenotypes are: "answer": ["B"]"* | The compressed representation is sufficient when the model preserves both clinical reasoning coherence and task-format instructions. |
| Failure Mode A: reasoning error with format compliance | *"...Consider the patient's medical history, symptoms, and physical examination findings. The patient's vitals, lab results, and medical history suggest that the patient may have Acute and unspecified renal failure (A) and Chronic kidney disease (infection)(X). Therefore, the prediction should be: "answer": ["V"]"* | The model remains machine-evaluable, but the reasoning trajectory drifts away from the clinical evidence, leading to an incorrect label. |
| Failure Mode B: format compliance breakdown | *"The patient's clinical events suggest the patient had a renal failure (A) and cardiac dysrhythmias (B)... The patient also had a history of chronic kidney disease (H) and chronic kidney disease (E) and bronchiectasis (F)... Therefore, the patient is most likely to have (A) renal failure, and B) myocardial infarction (C) upon discharge."* (Note: Missing the "answer": [...] entirely) | The model loses the instruction-following context required for automated evaluation, even when parts of the clinical reasoning remain plausible. |

*Table 11.* Top-10 most frequent TPE tokens in MIMIC-IV. Counts represent frequencies observed across the corpus.

| | Gemma2 | | Llama3 / Qwen2.5 | |
|---|---|---|---|---|
| Rank | Token | Count | Token | Count |
| 1 | Infusion | 387,560 | mmH | 672,434 |
| 2 | /min | 308,489 | Infusion | 387,560 |
| 3 | /L | 273,205 | olic blood | 368,487 |
| 4 | terial | 186,966 | Nonin | 317,616 |
| 5 | Moles | 95,253 | pressure by | 317,395 |
| 6 | Procedure: | 56,901 | vasive is | 317,395 |
| 7 | #/volume | 42,979 | insp/min | 239,428 |
| 8 | Alarms | 36,111 | pressure is | 235,231 |
| 9 | Arter | 32,942 | Oxygen | 220,037 |
| 10 | CVP | 29,220 | Heart rate | 180,794 |

dataset degrades performance on both tasks. This decline is driven by differences in term frequencies and lexical distributions across distinct healthcare scenarios (Hu et al., 2026). As indicated by the drops of CR, the highly frequent $N$-grams mined from MIMIC-IV appear less frequently in EHRSHOT patient records. Consequently, the model is forced to rely on misaligned sub-token representations. Furthermore, the results show that Llama3-1B is highly sensitive to this domain shift while Qwen2.5-1.5B demonstrates relative transfer robustness. This disparity can be attributed to the ability of higher-capacity base models to partially mitigate the impact of out-of-domain token representations (Calderon et al., 2024). However, while robust LLMs can partially absorb the shock of out-of-domain token representations, achieving optimal efficiency and predictive performance requires domain-specific vocabulary mining and SSFT to capture the unique linguistic distributions.

## J. Context-length Robustness Evaluation

Appendix Figure 5 further illustrates the context-length robustness of MedTPE across a diverse set of LLMs and clinical prediction tasks. Across all model sizes and both MIMIC-IV and EHRSHOT benchmarks, MedTPE consistently matches or outperforms the original tokeniser at varying input lengths. This pattern holds for both small and large models, including Llama3-1B, Qwen2.5-1.5B, Qwen2.5-7B, Llama3-8B, and Meditron3-8B, and across multiple prediction tasks such as ICU mortality, phenotyping, 30-day readmission, and 1-year pancreatic cancer prediction. Although the magnitude of improvement varies by task and model, MedTPE demonstrates stable or enhanced F1 scores as the input window grows, confirming its effectiveness for long-context clinical inference. These results reinforce the conclusion that MedTPE is a robust solution for compressing clinical input sequences and enabling scalable LLM performance under extended context settings.

## K. Test-time scaling Evaluation

In this experiment, we adopt majority voting as the test-time scaling approach. In majority voting, the model generates multiple independent responses for each input, and the final prediction is determined by the most frequent output (Chen et al., 2024b). Figure 6 reports the improvement in the F1 score and the relative inference time for MedTPE, where the relative time is measured against the inference time of the original tokeniser for one round. The number of independent responses is denoted by $n$ ($n = 1, 3, 5$). In most LLMs and tasks on the MIMIC-IV and EHRSHOT

*Table 12.* Assessment of Llama3-1B on the MIMIC-IV Phenotyping task with baselines augmented with embedding fine-tuning (Emb-FT).

| Model | Phenotyping | | | |
|---|---|---|---|---|
| | F1 ↑ | FCR ↑ | Time ↓ | CR ↑ |
| Llama3-1B | $0.050_{\pm 0.002}$ | 0.492 | 50.2 | - |
| + T5Summary | $0.078_{\pm 0.003}$ | 0.838 | 13.6 | 98.9% |
| + T5Summary (Emb-FT) | $0.081_{\pm 0.004}$ | 0.810 | 13.2 | 98.9% |
| + LLMLingua2 | $0.058_{\pm 0.003}$ | 0.500 | 38.7 | 32.4% |
| + LLMLingua2 (Emb-FT) | $0.017_{\pm 0.001}$ | 0.469 | 34.9 | 32.4% |
| **+ MedTPE (Ours)** | $\mathbf{0.114}_{\pm 0.004}$ | **0.870** | **22.3** | **32.4%** |

*Table 13.* Evaluation of MedTPE cross-domain transferability. "In-Domain" refers to mining and SSFT on EHRSHOT. "MIMIC-IV → EHRSHOT" refers to applying the MIMIC-IV trained tokeniser to EHRSHOT.

| Model & Setup | 30-day Readmission | | | | 1-year Pancreatic Cancer | | | |
|---|---|---|---|---|---|---|---|---|
| | F1$_{\pm std}$↑ | FCR↑ | Time↓ | CR↑ | F1$_{\pm std}$↑ | FCR↑ | Time↓ | CR↑ |
| Llama3-1B | $0.098_{\pm 0.008}$ | 0.425 | 28.1 | - | $0.039_{\pm 0.006}$ | 0.748 | 28.5 | - |
| + In-Domain | $0.191_{\pm 0.011}$ | 0.877 | 16.6 | 27.5% | $0.047_{\pm 0.006}$ | 0.900 | 15.9 | 28.3% |
| + MIMIC-IV → EHRSHOT | $0.014_{\pm 0.004}$ | 0.360 | 25.8 | 17.8% | $0.012_{\pm 0.002}$ | 0.539 | 19.6 | 17.8% |
| Qwen2.5-1.5B | $0.205_{\pm 0.011}$ | 0.989 | 15.2 | - | $0.047_{\pm 0.010}$ | 0.877 | 30.8 | - |
| + In-Domain | $0.209_{\pm 0.011}$ | 0.990 | 10.0 | 22.8% | $0.066_{\pm 0.017}$ | 0.947 | 11.5 | 23.1% |
| + MIMIC-IV → EHRSHOT | $0.197_{\pm 0.012}$ | 0.989 | 13.8 | 13.1% | $0.031_{\pm 0.010}$ | 0.981 | 17.4 | 13.1% |

datasets, MedTPE enables improvements in the F1 score over the original tokeniser as the number of majority voting responses increases, without increasing relative inference time. This positive trend is observed for both small and large models, including Llama3-1B, Qwen2.5-1.5B, Qwen2.5-7B, Llama3-8B, and Meditron3-8B. These results demonstrate that MedTPE can be effectively integrated with test-time scaling strategies, delivering improved performance without incurring extra inference costs compared to original models.

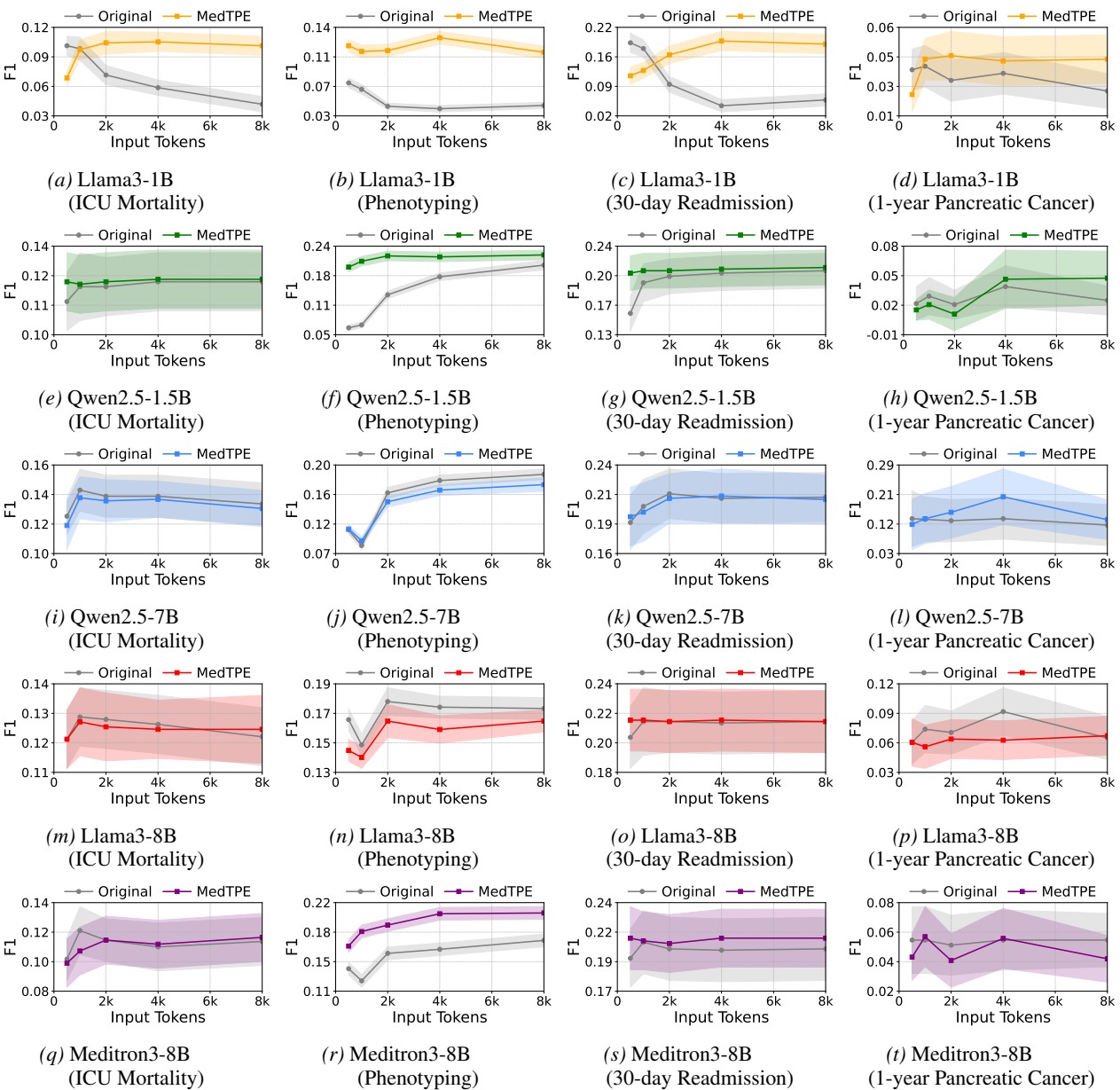

*Figure 5.* **Context-length robustness of MedTPE across LLMs and clinical tasks.** Each curve shows the mean F1 score (solid line for the original tokeniser, dashed line for MedTPE) with shaded areas indicating 95% confidence interval, evaluated across 0 to 8,192 input tokens. Token counts are measured using the original tokeniser for each LLM, ensuring that both models take the same amount of information. Subfigures (a–d) show results for Llama3-1B, (e–h) for Qwen2.5-1.5B, (i–l) for Qwen2.5-7B, (m–p) for Llama3-8B, and (q–t) for Meditron3-8B, each covering ICU mortality and phenotyping on MIMIC-IV, as well as 30-day readmission and 1-year pancreatic cancer prediction on EHRSHOT.

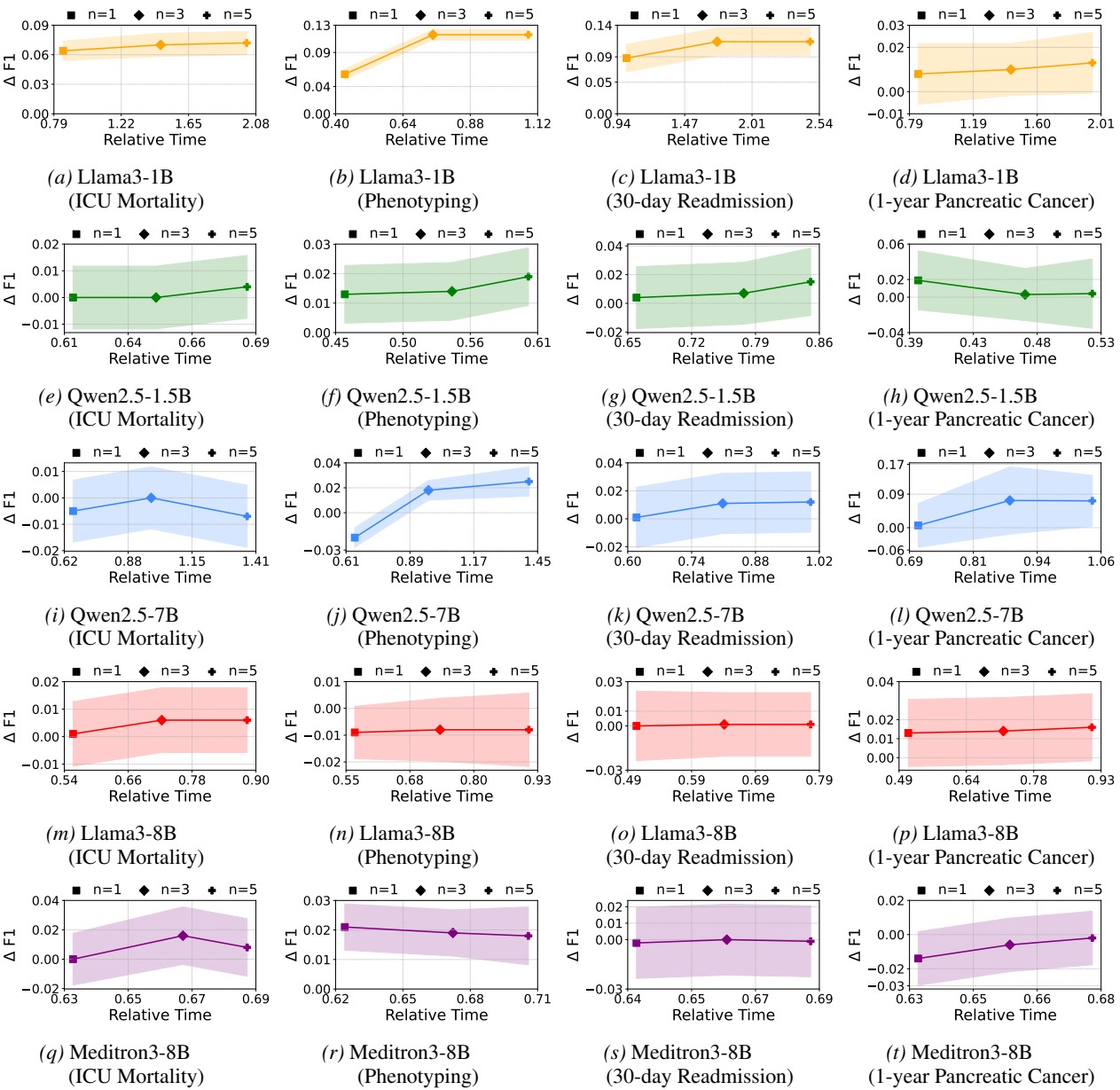

*Figure 6.* **Test-time scaling performance of MedTPE.** Each point shows the improvement in F1 score relative to the original tokeniser (y-axis) versus the relative inference time (x-axis) for different numbers of responses ($n = 1, 3, 5$). Subfigures (a–d) show results for Llama3-1B, (e–h) for Qwen2.5-1.5B, (i–l) for Qwen2.5-7B, (m–p) for Llama3-8B, and (q–t) for Meditron3-8B, each covering ICU mortality and phenotyping on MIMIC-IV, as well as 30-day readmission and 1-year pancreatic cancer prediction on EHRSHOT.

