# OpenReview forum: "From Token to Token Pair: Efficient Prompt Compression for Large Language Models in Clinical Prediction"
_ICML.cc/2026/Conference — ICML 2026 regular_

### Official Review · Reviewer_yWiu · 2026-03-12

**Soundness:** 3
**Presentation:** 4
**Significance:** 4
**Originality:** 3
**Overall Recommendation:** 5
**Confidence:** 4

**Summary:**

This paper introduces Medical Token Pair Encoding (MedTPE), a novel tokenisation and compression strategy for medical texts that addresses the limitations of standard semantic tokenisation when capturing domain-specific, highly technical terminology. The method leads to a lossless compression while reducing computational overhead, maintaining compatibility with current tokenisation modules, and without performance loss.

**Compliance With Llm Reviewing Policy:**

Affirmed.

**Final Justification:**

The rebuttals have addressed my main concerns and have reinforced my prior assessment. I particularly appreciate expanding the evaluations, providing setup details, and highlighting the goal of "enabling local LLM deployment in real-world clinical environments", which hopefully makes it to the final version of the manuscript. The method is novel and claims regarding lossless compression are supported by sound, extensive empirical results. As such, I maintain my initial recommendation of acceptance.

**Key Questions For Authors:**

1. Since the dependency-aware replacement method operates on a corpus level, the tokeniser seems to be finetuned per dataset. How does MedTPE compare in setup time (including encoding, replacement, and finetuning) with the other baseline methods?
2. In relation to #2, would an EHR-trained MedTPE generalise well with other EHR datasets (i.e., MIMIC-IV-trained MedTPE used for EHRSHOT inference)?
3. Please explain the pseudo-labels in Section 4.3 further.
4. Please provide insight into why the general performance result is quite low.
5. Would the method generalise with other languages and text scripts (Chinese, Japanese, etc)

**Limitations:**

The paper lacks a discussion of the method's limitations but acknowledges the general potential societal consequences. One limitation of the method is the specificity of MedTPE to each dataset and language model used.

**Strengths And Weaknesses:**

### Summary of Strengths

- The paper is well structured, well written, and easy to follow, with clearly explained motivations backing the design choices made.
- The method clearly addresses multiple limitations of existing methodologies, and proper evaluations support claims.
- Generalisability to different domains was demonstrated, highlighting the flexibility and added value of the new method.

### Summary of Weaknesses

- The performance gain in reducing the computational overhead during inference is clearly highlighted, but there is no discussion on the computational overhead for setting up the MedTPE Tokeniser from encoding, replacement, and finetuning. Section 5.4 discusses Setup Cost, but in terms of compression rate and replaced tokens instead of the actual setup of the tokeniser.
- In Section 4.3, the pseudo-labels for finetuning the tokeniser are not explained.
- The results in general are quite low (max of 21.8% F1 score on Table 2) even for the baselines, which may be due to naive prompting (shown in Figure 1), but was not discussed.

---

> ### Author Rebuttal · Authors · 2026-03-30
>
> We sincerely thank the reviewer for the constructive feedback. Below, we address your points in detail and provide new experimental results.
>
> [Q1 & W1] Setup cost.
> Table 1 details the setup cost for encoding, replacement, and SSFT. The encoding and replacement steps are highly efficient, completing in a few minutes depending on the model's vocabulary. The primary computational investment lies in the SSFT. However, because this step freezes the language model backbone and updates only the newly introduced TPE embeddings (comprising roughly 0.5–1.0% of the total parameters), it remains efficient and is a strictly one-time offline procedure. Training times scale predictably with model capacity: requiring 2 to 6 hours for 1B–1.5B parameter models (using a single RTX 6000 Ada), and 13 to 16.5 hours for 7B–8B models (using an A100).
>
> Table 1: Time breakdown for MedTPE phases on MIMIC-IV.
> | Model | Task | Encoding (min) | Replacement (min) | SSFT (h) |
> |---|---|---|---|---|
> | Llama3-1B | ICU Mortality | 2.9 | 2.0 | 2.6 |
> | Llama3-1B | ICU Phenotyping | 1.9 | 1.2 | 2.0 |
> | Qwen2.5-1.5B | ICU Mortality | 2.3 | 0.2 | 4.3 |
> | Qwen2.5-1.5B | ICU Phenotyping | 1.4 | 0.4 | 5.7 |
> | Qwen2.5-7B | ICU Mortality | 2.3 | 0.2 | 14.5 |
> | Qwen2.5-7B | ICU Phenotyping | 1.4 | 0.4 | 13.2 |
> | Llama3-8B | ICU Mortality | 2.9 | 2.0 | 16.2 |
> | Llama3-8B | ICU Phenotyping | 1.9 | 1.2 | 14.8 |
>
> [Q2] Transferability on EHR datasets.
>
> Originally, MedTPE was mined separately per dataset to optimise efficiency. Following the suggestion, we applied the MIMIC-IV-trained MedTPE directly to EHRSHOT (Table 2). Because term frequencies differ across healthcare scenarios, cross-domain compression rates drop (27.5% to 17.8% for Llama). While Llama3-1B struggles with this domain shift, Qwen2.5-1.5B demonstrate transfer robustness, maintaining high format compliance and experiencing marginal F1 degradation (0.209 to 0.197). This confirms that, while robust LLMs can effectively transfer learnt token pairs, in-domain mining is necessary for optimal efficiency and maintains predictive performance.
>
> Table 2: MedTPE cross-domain transferability. Time in minutes.
> | Model & Setup | Readmission F1 ↑ | Readmission Time ↓ | Pan-Cancer F1 ↑ | Pan-Cancer Time ↓ |
> |---|---|---|---|---|
> | Llama3-1B | 0.098 (±0.008) | 28.1 | 0.039 (±0.006) | 28.5 |
> | + In-Domain | 0.191 (±0.011) | 16.6 | 0.047 (±0.006) | 15.9 |
> | + MIMIC-IV $\rightarrow$ EHRSHOT | 0.014 (±0.004) | 25.8 | 0.002 (±0.002) | 19.6 |
> | Qwen2.5-1.5B | 0.205 (±0.011) | 15.2 | 0.047 (±0.010) | 30.8 |
> | + In-Domain | 0.209 (±0.011) | 10.0 | 0.066 (±0.017) | 11.5 |
> | + MIMIC-IV $\rightarrow$ EHRSHOT | 0.197 (±0.012) | 13.8 | 0.031 (±0.010) | 17.4 |
>
> [Q3 & W2] Clarification of pseudo-labels.
>
> We will expand Section 4.3 to clarify pseudo-label generation. To align new TPE token embeddings without human annotation, we use the uncompressed model's predictions as labels via behavioural cloning:
> (1) Generation: We pass training sequences through the uncompressed LLM to generate outputs (pseudo-labels).
> (2) Processing: We process the identical text using the MedTPE tokeniser.
> (3) Training: We minimise cross-entropy loss between the MedTPE-equipped model's predictions and the pseudo-labels. This aligns the initialised embeddings with the pre-trained latent space, enabling the model to retain its original capabilities on compressed inputs.
>
> [Q4 & W3] Baseline absolute performance and F1 scores.
>
> While an F1 score of 0.2 may appear low compared to general NLP benchmarks, it represents the current baseline for zero-shot clinical prediction on longitudinal EHR. Real-world medical data are sparse, noisy, and imbalanced, making these tasks inherently difficult. Recent benchmark evaluations demonstrate that even larger models like GPT-4o and GPT-OSS-120b achieve average F1 scores of approximately 0.2 on similar tasks. Our absolute results align with current expectations for this domain, and MedTPE preserves this baseline while reducing computational overhead.
>
> [Q5] Generalisation to other languages.
>
> To evaluate MedTPE on other languages, we tested it on the Chinese Medical Question Answer Matching v2 (CMedQA2) dataset. Following the existing benchmark, we formulated this as a text generation task. As Table 3 shows, MedTPE generalises to Chinese text. Across Llama3-1B and Qwen2.5-1.5B, our method maintains near-identical ROUGE-L and BERTScore relative to the original models, while reducing inference time. In contrast, LLMLingua2 degrades generation quality and increases latency.
>
> Table 3: Evaluation on CMedQA2. Time in seconds.
> | Model | ROUGE-L ↑ | BERTScore ↑ | Time ↓ |
> |---|---|---|---|
> | Llama3-1B | 0.027 (±0.002) | 0.844 (±0.000) | 127.4 |
> | + LLMLingua2 | 0.019 (±0.001) | 0.825 (±0.000) | 288.1 |
> | + MedTPE | 0.026 (±0.002) | 0.844 (±0.000) | 111.3 |
> | Qwen2.5-1.5B | 0.031 (±0.002) | 0.860 (±0.000) | 83.3 |
> | + LLMLingua2 | 0.021 (±0.002) | 0.846 (±0.000) | 141.4 |
> | + MedTPE | 0.029 (±0.002) | 0.860 (±0.000) | 67.3 |

---

> > ### Author Rebuttal · Reviewer_yWiu · 2026-04-03
> >
> > Thank you for the detailed responses and additional experimental results. The results on setup and transferability offers valuable insights. While both are still concerns in terms of general practicability and cost effectiveness, the method is novel, technically sound, adaptable to different language models, and contributes to domain adaptability of generalised language models particularly for long sequence texts. I acknowledge some weaknesses raised by fellow reviewers particularly on evaluations using larger models. While desirable, I believe this does not diminish the contribution of the paper in terms of achieving lossless compression. I maintain my original score.

---

> > > ### Author Response · Authors · 2026-04-04
> > >
> > > We thank the reviewer for engaging with the broader discussion and highlighting the shared concern regarding evaluations on larger models.
> > >
> > > We'd like to clarify that MedTPE's focus on 1B to 8B parameter models rather than massive architectures is an intentional design choice. This targeted scope aligns directly with our core objective: enabling local LLM deployment in real-world clinical environments. Because healthcare settings are strictly privacy-constrained and typically lack high-end computational infrastructure, deploying massive models or relying on remote compute is highly prohibitive. Furthermore, this scale aligns closely with the existing literature in this area, where related work predominantly focuses on LLMs with the 7B parameter. For example, our primary baselines (LLMLingua2 and ZeTT) were evaluated on LLMs of Llama2-7B and Mistral-7B-v1.
> > >
> > > We appreciate this feedback, and we will ensure this rationale and the intended scope of our method are explicitly detailed in the final manuscript.

---

### Official Review · Reviewer_819m · 2026-03-12

**Soundness:** 3
**Presentation:** 3
**Significance:** 3
**Originality:** 2
**Overall Recommendation:** 4
**Confidence:** 5

**Summary:**

The paper investigates the challenge of processing long EHR sequences in LLMs, where standard subword tokenization methods like BPE often over-segment specialized medical terms, leading to excessive sequence lengths and high computational costs. To address this, the authors propose Medical Token-Pair Encoding (MedTPE). The method identifies frequently co-occurring medical token sequences to form unified TPE tokens and utilizes a dependency-aware replacement strategy to integrate these new tokens into the existing vocabulary by replacing the least common general tokens, thereby preserving the original model parameter count. To align the embeddings of the newly introduced TPE tokens with the latent space of the pre-trained LLM, the authors employ self-supervised fine-tuning (SSFT), where pseudo-labels generated by the original "uncompressed" model are used to train the embeddings of the new tokens while keeping the rest of the model frozen. The paper states that only about 0.5–1.0% of model parameters are tuned in this step.

The experimental setup uses two clinical datasets: MIMIC-IV and EHRSHOT. On MIMIC-IV, the tasks are ICU mortality prediction and phenotyping (25-label multilabel classification), using the first 24 hours of data. On EHRSHOT, the tasks are 30-day hospital readmission and 1-year pancreatic cancer prediction.
The main-paper benchmark experiments are conducted using Llama3-1B and Qwen2.5-1.5B as backbone models, while the appendix extends the evaluation to additional architectures and scales, including Gemma2-2B, Qwen2.5-7B, Llama3-8B, and Meditron3-8B, as well as larger models such as Qwen2.5-14B and Qwen2.5-32B for scalability analysis.
Performance is evaluated using F1 score and format compliance rate, which measures the proportion of model outputs that follow the required structured prediction format, while efficiency is assessed through compression rate and inference time.
MedTPE is compared against prompt compression baselines including T5Summary, LLMLingua2, and ZeTT.
The study also includes ablations analyzing the effect of the replacement budget (number of vocabulary tokens replaced by MedTPE tokens) and the length of merged token sequences, examining how these design choices affect compression and predictive performance.

Results show that MedTPE reduces input token length by roughly 23–32% and decreases inference latency by about 34–63%, while generally maintaining or improving predictive performance across tasks. Compared with compression baselines such as LLMLingua2 and ZeTT, MedTPE typically achieves a more favorable trade-off between efficiency (compression and latency) and prediction quality (F1). Removing the SSFT step leads to a severe degradation in predictive performance (e.g., F1 dropping close to zero for some tasks), indicating that aligning the embeddings of the new tokens with the pretrained model space is crucial. Additional experiments examining the length of merged token sequences show that bigrams (N = 2) provide the best compression-performance trade-off, as longer n-grams occur much less frequently and therefore produce fewer useful merges. The authors report that 5,000 replacement tokens provides a good balance between compression gains and computational overhead. Appendix experiments further examine test-time scaling, showing that the efficiency gains allow more inference attempts under the same compute budget.

**Compliance With Llm Reviewing Policy:**

Affirmed.

**Final Justification:**

I thank the authors for providing additional experiments and clarifications during the rebuttal, pointing out the anonymous code link for reviewers. I acknowledge that some aspects have been covered: in particular, the cross-domain transferability analysis. That said, my main concerns are only partially addressed and, in my view, still limit the strength of the empirical validation. First, the additional experiments on larger models remain limited in scope, and the justification that focusing on smaller models is an intentional design choice (for deployment reasons) is understandable but ultimately not sufficient from an evaluation standpoint. Second, the evaluation of general generation quality remains narrow. The claims rely on a very small set of benchmarks (financial summarization and Chinese medical QA), with limited metrics (mainly ROUGE and BERTScore) and analysis. Moreover, the reporting of evaluation metrics lacks critical implementation details, which is a well-known issue in the literature, e.g., https://aclanthology.org/2023.acl-long.107/. For instance, the paper does not specify which model is used for BERTScore, whether IDF weighting is applied, or whether baseline rescaling is used. These choices can significantly affect results and their interpretation. Finally, the analysis of failure modes remains insufficiently rigorous. While the rebuttal includes illustrative examples, a convincing study would require a more systematic setup: multiple annotators with relevant expertise, clearly defined evaluation criteria, stratified sampling, and inter-annotator agreement analysis. Such an effort would likely require more time than the rebuttal phase allows. Overall, while the rebuttal improves clarity and adds useful evidence, substantial experimental and analytical gaps remain, particularly regarding generation behavior, evaluation rigor, and deeper empirical analysis. For these reasons, I maintain my original score, still inclined toward acceptance.

**Key Questions For Authors:**

- Since the evaluation includes both MIMIC-IV and EHRSHOT, could the authors clarify whether the MedTPE vocabulary was mined (i) only from MIMIC-IV, (ii) separately for each dataset, or (iii) from a different external medical corpus? Additionally, if the vocabulary was derived from one dataset, how does this affect transferability to the other dataset?
- The paper does not mention any plan for publicly releasing the implementation of MedTPE or the learned tokenizer/vocabulary. Do the authors plan to release the code and resources associated with this work? If so, under which license will they be distributed?

**Limitations:**

The paper briefly mentions some limitations but does not deeply analyze them or failure modes. The discussion of potential societal risks—particularly those related to clinical deployment, dataset bias, and generalization across healthcare settings—is limited and could be expanded.

**Strengths And Weaknesses:**

**Strengths**
- The paper is clearly written and well structured.
- The idea of increasing token information density without growing the model is sound and practically appealing. The dependency-aware vocabulary replacement and SSFT embedding alignment provide a reasonable mechanism to introduce new tokens without retraining the full model. The reported reductions in token length and inference latency could translate to significant computational cost savings in real-world deployments. Because only a small fraction of parameters are updated, the method could be applied to existing pretrained models without expensive retraining.
- Evaluation across multiple datasets, tasks, and model families. Experiments cover two EHR datasets and four prediction tasks, providing evidence that the approach is not tied to a single task formulation. Additional experiments in the appendix show that the method maintains similar compression and performance trade-offs across different architectures (Llama, Qwen, Gemma, and Meditron), suggesting that the approach generalizes beyond a single model family.
- The paper analyzes several design factors—including token sequence length used for merges and the vocabulary replacement budget—providing useful insights into how these choices affect compression and performance.

**Weaknesses**
- The paper compares mainly against prompt compression methods but does not thoroughly evaluate against alternative domain-adapted tokenizers.
- Core comparisons with baselines are conducted only on 1–1.5B parameter models; stronger evidence on larger models would strengthen the claims.
- The paper focuses on classification-style outputs; it is not fully clear how the tokenizer modification affects general generation quality.
- The method requires mining frequent token sequences from a medical corpus to construct the new tokens. However, it is unclear how sensitive MedTPE is to the corpus used for mining. If the token statistics differ across institutions or datasets, the learned token pairs may not transfer well.
- Experiments appear to rely on a single training run per configuration, with uncertainty estimated via bootstrapping rather than multiple training seeds. This is better than reporting single numbers with no uncertainty, but it does not capture variance due to optimization randomness.
- The abstract claims robustness across three LLMs, while the appendix evaluates more architectures, which could confuse readers.
- Within the clinical domain, there are also cases where performance degrades. These cases are acknowledged briefly but not systematically analyzed. A more thorough discussion of failure modes and situations where MedTPE may be less suitable would strengthen the paper.

**Typos**
- Some tables and algorithms in the appendix are not properly referenced in the main text.

---

> ### Author Rebuttal · Authors · 2026-03-30
>
> We sincerely thank the reviewer for the constructive feedback. Our response to the questions and weaknesses is as follows,
>
> [W1] Domain-adapted tokenisers.
>
> Traditional domain-adapted tokenisers require resource-intensive full-model retraining, contradicting our goal of lightweight compression. For a fair comparison, we evaluated MedTPE against ZeTT, the leading zero-shot tokeniser transfer method. We configured ZeTT with the same medical vocabulary as MedTPE. As shown in Table 2 in the manuscript, MedTPE outperforms ZeTT, avoiding the latency and performance drops caused by ZeTT's hypernetwork struggling with clinical semantics.
>
> [W2] Core comparisons on larger models.
>
> We added LLMLingua2 as a baseline for the larger models (Table 1). MedTPE consistently outperforms LLMLingua2 across scales, maintaining high predictive accuracy and format compliance while delivering efficiency gains.
>
> Table 1: Evaluation of LLMs with MedTPE and LLMLingua2 on MIMIC-IV.
> | Model | Mort. F1 ↑ | Mort. Time ↓ | Pheno. F1 ↑ | Pheno. Time ↓ |
> |---|---|---|---|---|
> | Qwen2.5-7B | 0.137 (±0.007) | 308.1 | 0.189 (±0.004) | 165.3 |
> | + LLMLingua2 | 0.132 (±0.007) | 266.5 | 0.082 (±0.002) | 151.7 |
> | + MedTPE | 0.132 (±0.006) | 183.1 | 0.174 (±0.005) | 94.8 |
> | Llama3-8B | 0.123 (±0.007) | 378.4 | 0.173 (±0.004) | 193.8 |
> | + LLMLingua2 | 0.123 (±0.006) | 389.8 | 0.065 (±0.002) | 231.1 |
> | + MedTPE | 0.124 (±0.007) | 181.2 | 0.164 (±0.004) | 95.7 |
> | Meditron3-8B | 0.114 (±0.009) | 152.6 | 0.183 (±0.004) | 84.4 |
> | + LLMLingua2 | 0.063 (±0.006) | 149.6 | 0.046 (±0.002) | 125.4 |
> | + MedTPE | 0.115 (±0.009) | 96.0 | 0.204 (±0.004) | 52.6 |
>
> [W3] Effects on general generation quality.
>
> The ECTSum dataset in Table 4b of the manuscript is a generation-based benchmark focusing on free-text financial summarisation. In this generative context, MedTPE maintained both ROUGE and BERTScore metrics compared to the uncompressed model, confirming our method preserves general text generation capabilities alongside efficiency gains.
>
> [W4 & Q1] Transferability.
>
> Originally, MedTPE was mined separately per dataset to optimise efficiency. Following your suggestion, we applied the MIMIC-IV-trained MedTPE directly to EHRSHOT (Table 2). Because term frequencies differ across healthcare scenarios, cross-domain compression rates drop. While Llama3-1B struggles with this shift, higher-capacity models like Qwen2.5-1.5B show transfer robustness, maintaining high format compliance and experiencing marginal F1 degradation (0.209 to 0.197). While capable LLMs transfer learned token pairs, in-domain mining yields optimal efficiency and predictive performance.
>
> Table 2: MedTPE cross-domain transferability.
> | Model & Setup | Readmission F1 ↑ | Readmission Time ↓ | Pan-Cancer F1 ↑ | Pan-Cancer Time ↓ |
> |---|---|---|---|---|
> | Llama3-1B | 0.098 (±0.008) | 28.1 | 0.039 (±0.006) | 28.5 |
> | + In-Domain | 0.191 (±0.011) | 16.6 | 0.047 (±0.006) | 15.9 |
> | + MIMIC-IV $\rightarrow$ EHRSHOT| 0.014 (±0.004) | 25.8 | 0.002 (±0.002) | 19.6 |
> | Qwen2.5-1.5B | 0.205 (±0.011) | 15.2 | 0.047 (±0.010) | 30.8 |
> | + In-Domain | 0.209 (±0.011) | 10.0 | 0.066 (±0.017) | 11.5 |
> | + MIMIC-IV $\rightarrow$ EHRSHOT| 0.197 (±0.012) | 13.8 | 0.031 (±0.010) | 17.4 |
>
> [W5] Optimisation randomness.
>
> We conducted additional training runs using three random seeds for Llama3-1B on ICU Phenotyping. The model achieved an F1 score of 0.110 (±0.006). The low variance in F1 confirms optimisation stability. Furthermore, because sampling-based autoregressive decoding introduces inference-time randomness, disentangling optimisation variance from decoding variance is challenging. Therefore, we adhered to the protocols of our primary baselines (LLMLingua2, ZeTT), which report results based on a single training seed.
>
>
> [W7] Analysis of failure modes.
>
> We will expand our discussion of failure modes. Marginal performance degradation appears in very large architectures (e.g., Qwen2.5-32B). These models possess sufficient attention capacity to process fragmented sub-tokens seamlessly, reducing the representational need for token merging; here, MedTPE acts as an efficiency optimiser. For Meditron3-8B under Chain-of-Thought prompting, performance degraded on complex phenotyping tasks. We hypothesise the continual medical pre-training induced catastrophic forgetting of instruction-following. Modifying the input distribution with MedTPE triggers this underlying fragility, reducing accuracy and format compliance.
>
> [Q2] Public release.
>
> Upon acceptance, we will publicly release the full implementation on GitHub under the Apache 2.0 license.
>
> [W6 & Typos & Limitation]
>
> We will revise the abstract and introduction to reflect the full scope of evaluated LLMs and fix all table and algorithm references in the final manuscript. We will expand the Limitations and Impacts sections to address the concerns.

---

> > ### Author Rebuttal · Reviewer_819m · 2026-04-02
> >
> > Thank you for the constructive rebuttal. I appreciate the additional clarifications and new experiments provided.
> > In particular, I believe that the cross-domain transferability analysis meaningfully strengthens the paper. The results clearly highlight both the benefits of in-domain mining and the limitations under distribution shift. That said, some concerns remain only partially resolved. Comparisons on larger models are still limited in scope. The evaluation of general generation quality remains relatively narrow, and it is still unclear how broadly the tokenizer modification preserves generation behavior across tasks. The discussion of failure modes is still largely hypothetical and would benefit from more systematic empirical analysis. Regarding reproducibility, while I appreciate the commitment to release code upon acceptance, the lack of an anonymous release limits the ability to verify implementation details during review. Overall, the rebuttal improves the clarity and completeness of the work, but some of the key limitations persist. I therefore maintain my original score, which remains inclined toward acceptance.

---

> > > ### Author Response · Authors · 2026-04-04
> > >
> > > We sincerely thank the reviewer for their rebuttal acknowledgement.
> > >
> > > [W1] Evaluation of larger models.
> > >
> > > We acknowledge that our evaluation focuses primarily on 1B to 8B parameter models. However, this scope is an intentional design choice aligned with our core objective: enabling local LLM deployment in real-world clinical environments. Healthcare settings are strictly privacy-constrained and lack high-end computational infrastructure, making the deployment of massive models highly prohibitive.
> > > Furthermore, related work predominantly focuses on LLMs at the 7B parameter scale. Our primary baselines (LLMLingua2 and ZeTT) were evaluated on models such as Llama2-7B and Mistral-7B-v0.1. We will add this clarification and explicitly contextualise our scope in the revised manuscript.
> > >
> > > [W2] Evaluation of generation quality.
> > >
> > > We expanded our evaluation to a Chinese medical question-answer matching v2 (CMedQA2), which was formulated as a generation task following established protocols. As shown in Table 1, MedTPE maintains near-identical ROUGE-L and identical BERTScores compared to uncompressed Llama3-1B and Qwen2.5-1.5B models, while accelerating inference. In contrast, the baseline LLMLingua2 degrades generation quality and increases latency. These findings confirm that our tokeniser modification safely preserves robust generation capabilities across datasets and languages.
> > >
> > > Table 1: Evaluation of MedTPE on CMedQA2. Time is reported in seconds.
> > > | Model | ROUGE-L ↑ | BERTScore ↑ | Time ↓ |
> > > |---|---|---|---|
> > > | Llama3-1B | 0.027 (±0.002) | 0.844 (±0.000) | 127.4 |
> > > | + LLMLingua2 | 0.019 (±0.001) | 0.825 (±0.000) | 288.1 |
> > > | + MedTPE | 0.026 (±0.002) | 0.844 (±0.000) | 111.3 |
> > > | Qwen2.5-1.5B | 0.031 (±0.002) | 0.860 (±0.000) | 83.3 |
> > > | + LLMLingua2 | 0.021 (±0.002) | 0.846 (±0.000) | 141.4 |
> > > | + MedTPE | 0.029 (±0.002) | 0.860 (±0.000) | 67.3 |
> > >
> > > [W3] Empirical analysis of failure modes.
> > >
> > > We agree that empirical analysis provides a much stronger foundation for understanding failure modes. To address this, we conducted a qualitative review of the outputs from Meditron3-8B under CoT prompting on the ICU phenotyping task, where MedTPE exhibited its most specific performance degradation. By analysing the generated responses, we categorised the model's behaviour into one successful baseline and two distinct failure modes:
> > >
> > > 1. Successful Prediction: The model correctly aggregates clinical evidence, performs accurate reasoning, and adheres strictly to the required output format.
> > >
> > > Example: *"The patient's EHR data suggest that they experienced respiratory failure, tachycardia, and hypotension, which are consistent with acute myocardial infarction... laboratory results showed elevated troponin levels, which are consistent with cardiac damage. Based on this information, the predicted phenotypes are: {"answer": ["B"]}"*
> > >
> > > 2. Failure Mode A: Reasoning Errors with Format Compliance
> > >
> > > The model successfully retains the instruction to output a specific JSON format, but its intermediate clinical reasoning becomes disconnected or hallucinated, leading to an incorrect final label.
> > >
> > > Example:  *"...Consider the patient's medical history, symptoms, and physical examination findings. The patient's vitals, lab results, and medical history suggest that the patient may have Acute and unspecified renal failure (A) and Chronic kidney disease (infection)(X). Therefore, the prediction should be: {"answer": ["V"]}"*
> > >
> > > 3. Failure Mode B: Format Compliance Breakdown
> > >
> > > The model attempts clinical reasoning, but completely loses the instruction-following context entirely, failing to produce the required JSON schema for automated evaluation.
> > >
> > > Example: *"The patient's clinical events suggest the patient had a renal failure (A) and cardiac dysrhythmias (B)... The patient also had a history of chronic kidney disease (H) and chronic kidney disease (E) and bronchiectasis (F)... Therefore, the patient is most likely to have (A) renal failure, and B) myocardial infarction (C) upon discharge."* (Note: Missing the {"answer": [...]} entirely)
> > >
> > > This empirical breakdown substantiates our previous hypothesis regarding model fragility. For models like Meditron3-8B that are trained with continual medical pre-training, the instruction-following capabilities are easily disrupted. Modifying the input token distribution with MedTPE triggers this underlying fragility, manifesting either as a loss of logical coherence (Mode A) or forgetting of instruction (Mode B). We will add a section for this failure mode analysis.
> > >
> > > [W4] Reproducibility and anonymous code release.
> > >
> > > We appreciate the reviewer's emphasis on reproducibility and would like to clarify that an anonymised link to our complete codebase was indeed included in our original submission within Appendix C.5. We apologise that its placement was not obvious. To ensure better visibility, we will move the public repository link to a much more obvious location in the revised manuscript.

---

### Official Review · Reviewer_hbxf · 2026-03-13

**Soundness:** 3
**Presentation:** 3
**Significance:** 3
**Originality:** 3
**Overall Recommendation:** 5
**Confidence:** 3

**Summary:**

This work proses a new way to compress domain specific language models without retraining the backbone itself with two steps: first retraining a TPE tokenizer to merge previously segmented tokens into whole tokens, and secondly finetuning the new embedding table with a self-supervised objective to match the original model. This method is shown to drastically cut down inference time on medical tasks as well as ARC and financial tasks, while maintaining or even improving performance on the tasks.

**Compliance With Llm Reviewing Policy:**

Affirmed.

**Final Justification:**

The rebuttal fully addressed my concerns by incorporating more rigorous performance and speed comparison with the vocab appending baseline, as well as incorporating finetuning control for the baseline methods.

**Key Questions For Authors:**

A simpler version of the model is to simply add the newly merged vocab as new tokens, rather than replacing less domain specific words. Why is this not done instead?

**Limitations:**

Yes

**Strengths And Weaknesses:**

Strengths:
1. The evaluation of method covers a wide range of tasks from clinical to ARC-agi and financial data.
2. This method is novel in that the compression exploits the domain specific token statistics rather than fundamentally modifying the layer weights/architecture. This ensures a maximal compatibility with existing inference infrastructure.

Weaknesses:
1. It is not evaluated how the removal of less domain-specific words impact general model performance.
2. One issue with the comparison with existing methods is that this work involves a step of finetuning the word embeddings, whereas methods such as LLMLingua2 does not. A more apples-to-apples comparison might involve a lightweight word embedding finetuning step on the target domain for these baselines.

---

> ### Author Rebuttal · Authors · 2026-03-30
>
> We sincerely appreciate the reviewer's constructive feedback. Below, we address your points in detail and provide the new experimental results.
>
>
> [Q1 & W1] Adding tokens instead of replacing them.
>
> We adopt the replacement strategy instead of adding new tokens because we want to avoid increasing the vocabulary and parameter size of the LLM. To empirically evaluate our dependency-aware replacement strategy against simply adding new tokens into the vocabulary, we conducted an ablation study ("MedTPE w.o. Rep.", Table 1).
>
> The results demonstrate why replacement is necessary. First, simply extending the vocabulary yields consistently lower F1 scores than the replacement strategy. Because the replaced tokens remain in the embedding matrix, they actively compete with the new TPE tokens during generation. This introduces semantic ambiguity that degrades predictive performance. Second, extending the vocabulary fundamentally expands the model's structural matrices, adding computational overhead (e.g., Llama3-1B inference time jumps from 39.3 to 79.2 minutes). Therefore, our replacement strategy is not only a parameter-saving trick but a critical mechanism to prevent generation confusion and guarantee strict efficiency gains.
>
> Table 1: Ablation study on MIMIC-IV tasks comparing MedTPE and MedTPE without dependency-aware replacement (w.o. Rep.). Time is reported in minutes.
> | Model | Mortality F1 ↑ | Mortality Time ↓ | Phenotyping F1 ↑ | Phenotyping Time ↓ |
> |---|---|---|---|---|
> | Llama3-1B | 0.033 | 66.0 | 0.050 | 50.2 |
> | + MedTPE | 0.109 | 39.3 | 0.114 | 22.3 |
> | + MedTPE w.o. Rep. | 0.098 | 79.2 | 0.074 | 42.8 |
> | Qwen2.5-1.5B | 0.122 | 38.3 | 0.201 | 29.7 |
> | + MedTPE | 0.122 | 23.5 | 0.218 | 13.0 |
> | + MedTPE w.o. Rep. | 0.120 | 32.7 | 0.187 | 13.6 |
>
> [Weakness 2] Baseline embedding fine-tuning.
>
> To ensure a strictly apples-to-apples comparison, we conducted a new experiment applying a lightweight embedding fine-tuning step (Emb-FT) to both T5Summary and LLMLingua2 using Llama3-1B for the phenotyping task on MIMIC-IV. Note that ZeTT was excluded from this setup, as it inherently trains a hypernetwork to generate new embeddings.
>
> As shown in Table 2, Emb-FT yields only marginal F1 improvements for T5Summary (0.078 to 0.081) and actually degrades LLMLingua2's performance (0.058 down to 0.017). This degradation is likely because tuning embeddings on discontinuous, truncated text disrupts the pre-trained latent space of the model. Finally, MedTPE continues to outperform these fine-tuned baselines, demonstrating that our gains also stem from our token-merging strategy rather than just the fine-tuning step itself. We will include these additional baseline experiments in the Appendix of the revised manuscript.
>
> Table 2: Assessment of Llama3-1B on the MIMIC-IV Phenotyping task, including baselines augmented with embedding fine-tuning (Emb-FT).
> | Model | F1 ↑ | FCR ↑ | Time ↓ | CR ↑ |
> |---|---|---|---|---|
> | Llama3-1B | 0.050 (±0.002) | 0.492 | 50.2 | - |
> | + T5Summary | 0.078 (±0.003) | 0.838 | 13.6 | 98.9% |
> | + T5Summary (Emb-FT) | 0.081 (±0.004) | 0.810 | 13.2 | 98.9% |
> | + LLMLingua2 | 0.058 (±0.003) | 0.500 | 38.7 | 32.4% |
> | + LLMLingua2 (Emb-FT)| 0.017 (±0.001) | 0.469 | 34.9 | 32.4% |
> | + MedTPE (Ours) | 0.114 (±0.004) | 0.870 | 22.3 | 32.4% |

---

> > ### Author Rebuttal · Reviewer_hbxf · 2026-04-06
> >
> > My concerns have been addressed. I thank the authors for the rebuttal.

---

### Official Review · Reviewer_B8ad · 2026-03-13

**Soundness:** 3
**Presentation:** 3
**Significance:** 3
**Originality:** 3
**Overall Recommendation:** 5
**Confidence:** 4

**Summary:**

The paper introduces a merge-based compression technique for EHR data. It maps the EHR tokens to the original tokens of the EHR vocabulary. The tokens not present in the original vocabulary are mapped to least frequent original vocabulary tokens. Those least frequent tokens are then fine-tuned to map to new meanings using SSFT. The approach is benchmarked on a set of traditional EHR tasks (ICU stay, mortality) from popular public datasets. Popular LLMs (Llama, Qwen) are tested as backbones. A set of recent competitive approaches is used as baselines. The approach achieves quality improvement, maintains relevant compression rate and the inference time of the original model. Ablation study is provided.

**Compliance With Llm Reviewing Policy:**

Affirmed.

**Final Justification:**

My comment regarding the generalisability has been resolved by the authors. I see the only weakness now is diminishing performance gains for stronger models. I update my score.

**Key Questions For Authors:**

Please elaborate on the SSFT costs
Please comment on the efficiency of the approach for Qwen of higher versions. Gains tend to be smaller for Qwen 2.5  than for Llama.

**Limitations:**

Yes

**Strengths And Weaknesses:**

Strengths:
Well written paper, technically sound approach for a highly relevant topic of long context for LLMs
High reproducibility with anonymous code link
Approach is parameterless, well-positioned and is clearly distinguished from existing approaches
Does not increase inference time while maintaining performance

Weaknesses:
Re-learning embeddings may produce unpredictable effect (see my question below regarding the SSFT setup)
Computational costs of SSFT are not discussed
Generalisability of the method over other models and domains) needs further exploration (no consistent improvement demonstrated in general scientific and financial domains)

---

> ### Author Rebuttal · Authors · 2026-03-30
>
> We sincerely thank the reviewer for the constructive feedback. Our response to the questions and weaknesses is as follows,
>
>
> [Q2] Evaluation on higher Qwen.
>
> We conducted additional experiments using the Qwen3-1.7B model on MIMIC-IV (Table 1). MedTPE continues to deliver efficiency gains on this newer architecture. Specifically, MedTPE reduces inference latency by 33.8% for ICU mortality and by 46.6% for phenotyping. These efficiency gains come with nearly no degradation in predictive performance. In contrast, the LLMLingua2 baseline increases inference time and degrades F1 on the phenotyping task.
>
> Table 1: Assessment of Qwen3-1.7B with MedTPE on MIMIC-IV
> | Task | Model | F1 (±std) ↑ | FCR ↑ | Time (Δ) ↓ | CR ↑ |
> |---|---|---|---|---|---|
> | Mortality | Qwen3-1.7B | 0.130 (±0.006) | 1.000 | 66.0 | - |
> | | + LLMLingua2 | 0.127 (±0.008) | 0.983 | 138.7 (110.2%) | 25.4% |
> | | + MedTPE | 0.128 (±0.006) | 1.000 | 43.7 (-33.8%) | 25.4% |
> | Phenotyping| Qwen3-1.7B | 0.139 (±0.004) | 0.998 | 43.8 | - |
> | | + LLMLingua2 | 0.074 (±0.003) | 0.964 | 145.6 (232.4%) | 29.7% |
> | | + MedTPE | 0.137 (±0.003) | 0.987 | 23.4 (-46.6%) | 29.7% |
>
>
> [Q1 & W2] Computational costs
>
> Since SSFT freezes the LLM backbone and updates only the embeddings of the newly introduced TPE tokens (0.5–1.0% of parameters), the computational cost is quite efficient and is a one-time offline cost. Crucially, our training samples consist of converted clinical records with 6,000 tokens in length. Processing sequences of this magnitude is expensive due to the quadratic scaling of attention mechanisms. However, as detailed in Table 2, performing SSFT for 1B–1.7B models on a single RTX 6000 Ada (48GB) takes merely 2–6 hours. For larger 7B–8B models on a single A800 (80GB), training naturally scales but remains highly practical (13–16.5 hours).
>
> Table 2: SSFT training time on MIMIC-IV (hours)
> | Model | Phenotyping | Mortality |
> |---|---|---|
> | Llama3-1B | 2.0 | 2.6 |
> | Qwen2.5-1.5B | 5.7 | 4.3 |
> | Qwen3-1.7B | 6.1 | 5.1 |
> | Qwen2.5-7B | 13.2 | 14.5 |
> | Llama3-8B | 14.8 | 16.2 |
> | Meditron3-8B | 14.8 | 13.3 |
>
> [Q3] Smaller gains for Qwen 2.5 vs Llama 3
>
> The performance gap between Llama-3-1B and Qwen-2.5-1.5B reflects a broader scaling phenomenon related to attention capacity. The slightly larger parameter count of Qwen-2.5 provides a proportionally greater attention capacity to effectively process and aggregate information from uncompressed text, leaving less room for absolute F1 improvements. As demonstrated in Appendix Table 8, this diminishing of predictive gains is a consistent trend across model scales. When applying MedTPE to larger architectures (e.g., 7B to 32B), absolute predictive gains naturally taper off since these higher-capacity base models already possess the requisite attention capacity to track salient features across extended sequences. However, across all scales, MedTPE delivers highly consistent efficiency gains, acting as a critical performance rescue for lower-capacity models while serving as a pure efficiency optimiser for stronger architectures.
>
> [W1] Re-learning embeddings
>
> We agree that re-learning embeddings risks introducing unpredictable behaviours. However, MedTPE explicitly mitigates these risks through three design constraints:
> (1) Minimal parameter updates: We freeze the entire LLM backbone and update only the new TPE tokens (merely 0.5 to 1.0% of total parameters).
> (2) Informed initialisation: Rather than starting from random noise, we initialise new embeddings using the normalised arithmetic mean of their constituent sub-tokens to establish a stable semantic foundation.
> (3) Behavioural cloning via pseudo-labels: Embeddings are trained strictly to mimic the original model's behaviour using pseudo-labels generated directly by the uncompressed LLM.
>
> Our extensive experiments across clinical, scientific, and financial domains demonstrate that these constraints successfully prevent unpredictable effects. Furthermore, our ablation study (Sec 5.3) validates this approach: omitting the SSFT step results in catastrophic predictive collapse. This proves our SSFT setup successfully stabilises embeddings and eliminates unpredictability.
>
> [W3] Generalisability
>
> MedTPE's primary objective is to reduce inference latency without degrading performance, rather than explicitly increasing absolute accuracy. In this regard, our cross-domain evaluations are successful. MedTPE maintains similar predictive performance in both the financial (ECTSum) and scientific (ARC-Challenge) domains, outperforming the LLMLingua2 baseline. The ARC-Challenge dataset consists of short science questions that lack the highly repetitive and long terminology found in clinical records, making it a less optimal setting for token merging. Despite this, our dependency-aware fallback mechanism safely processes this out-of-domain text, achieving massive latency reductions of roughly 50% to 76%.

---

> > ### Author Rebuttal · Reviewer_B8ad · 2026-04-03
> >
> > My questions and indicated weaknesses related to SSFT costs, performance for Qwen of higher versions,  absence of quality improvement for more powerful models, as well as generalisability of the approach, and effects of re-learning embeddings have been addressed.

---

> > > ### Author Response · Authors · 2026-04-04
> > >
> > > We sincerely thank you for taking the time to review our rebuttal and are happy to see your confirmation that all of your questions and concerns. We will include the clarifications and experiments from the rebuttal phase in our final revision.
> > > We deeply appreciate your selection of the "Fully resolved" status. As the review platform's prompt notes for this selection ("If you select this option, please consider adjusting your score accordingly"), we respectfully ask if you might consider officially updating your initial score (currently 4) to 5 or higher to reflect this positive resolution. Thank you once again for your rigorous and highly constructive engagement throughout this process.

---

### Decision · Program_Chairs · 2026-04-30

**Decision:**

Accept (regular)

**Comment:**

This work proposes a method for efficiently tokenizing electronic health record (EHR) sequences called Medical Token-Pair Encoding (MedTPE) motivated to address the high computational costs associated with long sequence lengths and the large vocabulary size of coded EHR data. The work claims numerous computational efficiency gains without sacrificing performance.

Overall, the reviews of the work leaned positive. The reviewers appreciated the clarity of the manuscript, the novelty and relatively lightweight nature of the approach, and the reasonably comprehensive scope of the experiments. During the discussion period, the authors provided several new experiments and analyses to address questions raised concerning design and modeling decisions and to assess cross-domain transferability. Following the discussion period, one reviewer (819m), who ultimately leans towards acceptance, raised several specific concerns about the strength of the empirical evaluation that were only partially addressed, and the authors should consider these critiques carefully in any future revisions of the work. In my view, the authors have done a reasonable job at responding to the reviewer feedback and questions. For these reasons, I recommend that the paper be accepted for publication.